# A resource of variant effect predictions of single nucleotide variants in model organisms

Omar Wagih[1], Marco Galardini[1], Bede P Busby[1,2] (ID), Danish Memon[1], Athanasios Typas[2] (ID) & Pedro Beltrao[1,*] (ID)

## Abstract

The effect of single nucleotide variants (SNVs) in coding and noncoding regions is of great interest in genetics. Although many computational methods aim to elucidate the effects of SNVs on cellular mechanisms, it is not straightforward to comprehensively cover different molecular effects. To address this, we compiled and benchmarked sequence and structure-based variant effect predictors and we computed the impact of nearly all possible amino acid and nucleotide variants in the reference genomes of *Homo sapiens*, *Saccharomyces cerevisiae* and *Escherichia coli*. Studied mechanisms include protein stability, interaction interfaces, post-translational modifications and transcription factor binding sites. We apply this resource to the study of natural and disease coding variants. We also show how variant effects can be aggregated to generate protein complex burden scores that uncover protein complex to phenotype associations based on a set of newly generated growth profiles of 93 sequenced *S. cerevisiae* strains in 43 conditions. This resource is available through mutfunc (www.mutfunc.com), a tool by which users can query precomputed predictions by providing amino acid or nucleotide-level variants.

**Keywords** burden score; genetic variants; genotype-to-phenotype; model organisms; resource

**Subject Categories** Chromatin, Epigenetics, Genomics & Functional Genomics; Computational Biology; Methods & Resources

**Mol Syst Biol. (2018) 14: e8430**

See also: **G Slodkowicz & M Madan Babu** (December 2018)

## Introduction

One of the key challenges of biology is to understand how genetic variation drives changes in phenotypes. Genome-wide association studies (GWASs) have made progress in identifying causal genetic loci, and over the past decade, a large number of associations have been made between genetic variation and phenotypic traits including disease risk (Welter *et al*, 2014). However, GWASs are typically limited in their ability to identify the causal variant at the associated locus and further limited by the ability to explain the underlying mechanism that may be influenced by candidate causal variants. This missing mechanistic layer severely limits our understanding of how variants cause phenotypic variability.

Variants occurring in coding and noncoding regions can influence a diversity of molecular functions. For instance, noncoding variants can affect chromatin accessibility (Kumasaka *et al*, 2016), splice sites (Xiong *et al*, 2015) and epigenetic modifications (Rintisch *et al*, 2014). Coding variants can affect post-translational modification (PTM) sites (Reimand *et al*, 2015; Wagih *et al*, 2015), protein folding and stability (Lorch *et al*, 2000), protein interaction interfaces (Engin *et al*, 2016) and subcellular localization (Björses *et al*, 2000), and introduce premature stop codons. Understanding the disrupted biological mechanisms underlying genetic variation is key to many applications in genetics such as genetically engineering organisms, assessing drug efficacy and drug discovery (Labaudinière, 2002; Lutz, 2010; Nelson *et al*, 2016).

The ability to predict the degree to which genetic variation would alter such mechanisms offers a time and cost-effective alternative over experimental approaches to prioritize variants of interest and to facilitate the understanding of the mechanisms underlying causal variants. A multitude of *in silico* predictors aimed at predicting such effects has been proposed (Schymkowitz *et al*, 2005; Kumar *et al*, 2009; Adzhubei *et al*, 2010; Wagih *et al*, 2015), yet they often require significant computational power, expertise and time to be used. Furthermore, each of the currently available tools does not comprehensively provide predicted effects across different molecular mechanisms (i.e. disruption of stability, interfaces, TF binding).

Accordingly, we have compiled and benchmarked commonly used sequence and structure-based predictors of mutational consequences and predicted the effect of nearly all possible variants in the reference genomes of *Homo sapiens*, *Saccharomyces cerevisiae* and *Escherichia coli*. The impact of variants was measured in the context of conserved protein regions, protein stability, protein–protein interaction (PPI) interfaces, PTMs, kinase–substrate interactions, short linear motifs (SLiMs), start and stop codons, and transcription factor (TF) binding sites (TFBSs). This resource is available through the mutfunc resource (http://mutfunc.com/),

1 European Molecular Biology Laboratory, European Bioinformatics Institute, Wellcome Trust Genome Campus, Cambridge, UK
2 European Molecular Biology Laboratory, Genome Biology Unit, Heidelberg, Germany
   *Corresponding author. Tel: +44 1223 494 610; E-mail: pbeltrao@ebi.ac.uk

which allows for prioritization of variants while providing insight into the altered mechanisms.

To demonstrate the utility of mutfunc, we assessed variants of uncertain clinical significance (VUSs) in *H. sapiens*. We further applied mutfunc to publically available variants for yeast *S. cerevisiae* strains to generate protein complex burden scores. We then phenotyped 93 sequenced *S. cerevisiae* strains in 43 conditions and utilized burden scores to associate protein complexes to phenotypes. This yielded associations that would not be possible through traditional variant-based GWAS approaches. mutfunc is a computational resource that will facilitate the study of the mechanistic impacts of genetic variation.

# Results

### Functional genomic regions display evolutionary constraint across yeast and human individuals

In order to set up the variant effect prediction approaches, we first derived, for *E. coli*, *S. cerevisiae* and *H. sapiens*, molecular information such as experimental and homology-based protein structural models for individual proteins and protein interfaces, TF binding sites, protein kinase targets sites, post-translational modification sites and linear motif regions (Materials and Methods). Structural models were used to identify interface residues and residues with different surface accessibility. Given that functionally relevant regions of the genome are under evolutionary constraint, we took the opportunity to use this large collection of functional regions to test whether these tend to be depleted of natural variants. For yeast, 896,772 natural variants and their allele frequencies were compiled from 405 yeast strains (Bergström *et al*, 2014; Strope *et al*, 2015; Gallone *et al*, 2016; Zhu *et al*, 2016), of which 478,857 were coding variants. For human, over 3.2M coding variants from over 65,000 individuals were obtained from the ExAC consortium (Lek *et al*, 2016).

Natural variants were mapped to 9,837 protein structures and homology models ($n$ = 6,737 human, $n$ = 3,100 yeast), and the residues were binned according to relative surface accessibility (RSA). Similarly, 9,883 structures ($n$ = 7,693 human, $n$ = 2,190 yeast) for protein interaction pairs were obtained from Interactome3D and the difference in surface accessibility (ΔRSA) between the unbound and bound complex was determined to identify interface residues, corresponding to those with the highest ΔRSA (Materials and Methods). The number of variants per position of each bin of RSA and ΔRSA was compared to counts observed in random positions in the protein, permuted 1,000 times. Fewer variants were found in buried regions and interface regions when compared to exposed regions in both yeast and human (Fig 1A $P < 1.28 \times 10^{-34}$ and B $P < 2.28 \times 10^{-33}$). To study variation at 296,147 and 26,560 human and yeast PTM sites, the variant counts over random expectation were calculated for a window of ±5 residues flanking the PTM positions. The level of constraint was different across PTM types (Fig 1C) with ubiquitylation showing the lowest level of constraint. Interestingly, the level of constraint for PTMs increases with the number of other neighbouring PTMs present in a 10 amino acid window (Fig 1D) suggesting that the clustering of PTMs may have important biological functions such as cross-talk regulation (Beltrao *et al*, 2013).

It has been shown that TF binding sites tend to have lower-than-expected variation across populations in particular for crucial specificity-determining positions (Spivakov *et al*, 2012). We therefore tested if similar observations are found at our putative TF binding sites for *S. cerevisiae* that were predicted using a combination of TF specificity models, TF knockout gene expression studies and TF ChIP-seq or ChIP-chip data (Materials and Methods). A total of 4,523 potential binding sites were identified across 93 TFs of *S. cerevisiae*. We computed the ratio between the variant counts within the predicted binding sites to that of random genomic sites of the same length and within the same ChIP regions. By combining the analysis across all putative binding sites of each TF, we observed that binding sites for some TFs are generally more constrained than others (Fig 1E). Those with higher levels of constraint include HAP4, a global regulator of respiratory genes and general transcriptional regulators such as REB1 and RAP1. At the level of individual TF binding sites, we observed that those found within clusters of binding sites tended to show higher levels of constraints than isolated sites (Fig 1F). Additionally, the TF binding positions for each TF were stratified according to their importance for binding as measured by the position-specific information content (IC) of the TF specificity position weight matrices. In accordance with expectation, positions with high IC, which correspond to positions that are important for binding, tend to have fewer variants than less important positions (Fig 1G). Position-specific constraint for individual TFs highlights this difference between high and low IC positions (Fig 1H).

Overall, these results provide an overview of how population-level variation differs across diverse set of genome functional elements and recapitulates findings from analysis of specific types of functional elements (Spivakov *et al*, 2012; de Beer *et al*, 2013; Reimand *et al*, 2015). Additionally, it suggests that our collection of functional elements (e.g. structures, interfaces, PTMs and TF binding sites) shows evolutionary constraints and therefore can be used further for the establishment of the variant effect prediction pipeline.

### A comprehensive resource of mechanistic effects of single nucleotide variants

We sought to better understand the mechanistic impact of point mutations affecting the above described functional elements. To do this, a set of commonly used predictors were used to assess the impact of every possible single amino acid or nucleotide substitution across *H. sapiens*, *S. cerevisiae* and *E. coli*, where applicable. We performed a large-scale computational estimation of the impact of variants on conserved protein regions, protein stability, protein interaction interfaces, kinase–substrate phosphorylation and other PTMs, linear motifs, TFBSs and start and stop codons (illustrated in Fig 2A, Materials and Methods). These results were deposited in the mutfunc resource, which offers a quick and interactive way by which users can gain predicted mechanistic insight for variants of interest. Although the algorithms used have been previously described, this resource allows to easily query all predictors in a unified and consistent interface.

To measure the impact on conserved regions, we constructed 29,027 multiple sequence alignments for proteins of the three organisms ($n$ = 19,497 *H. sapiens*, $n$ = 5,498 *S. cerevisiae*, $n$ = 4,032 *E. coli*) and used the SIFT algorithm (Ng & Henikoff, 2003) to assess the impact of all possible 291.7M protein coding variants

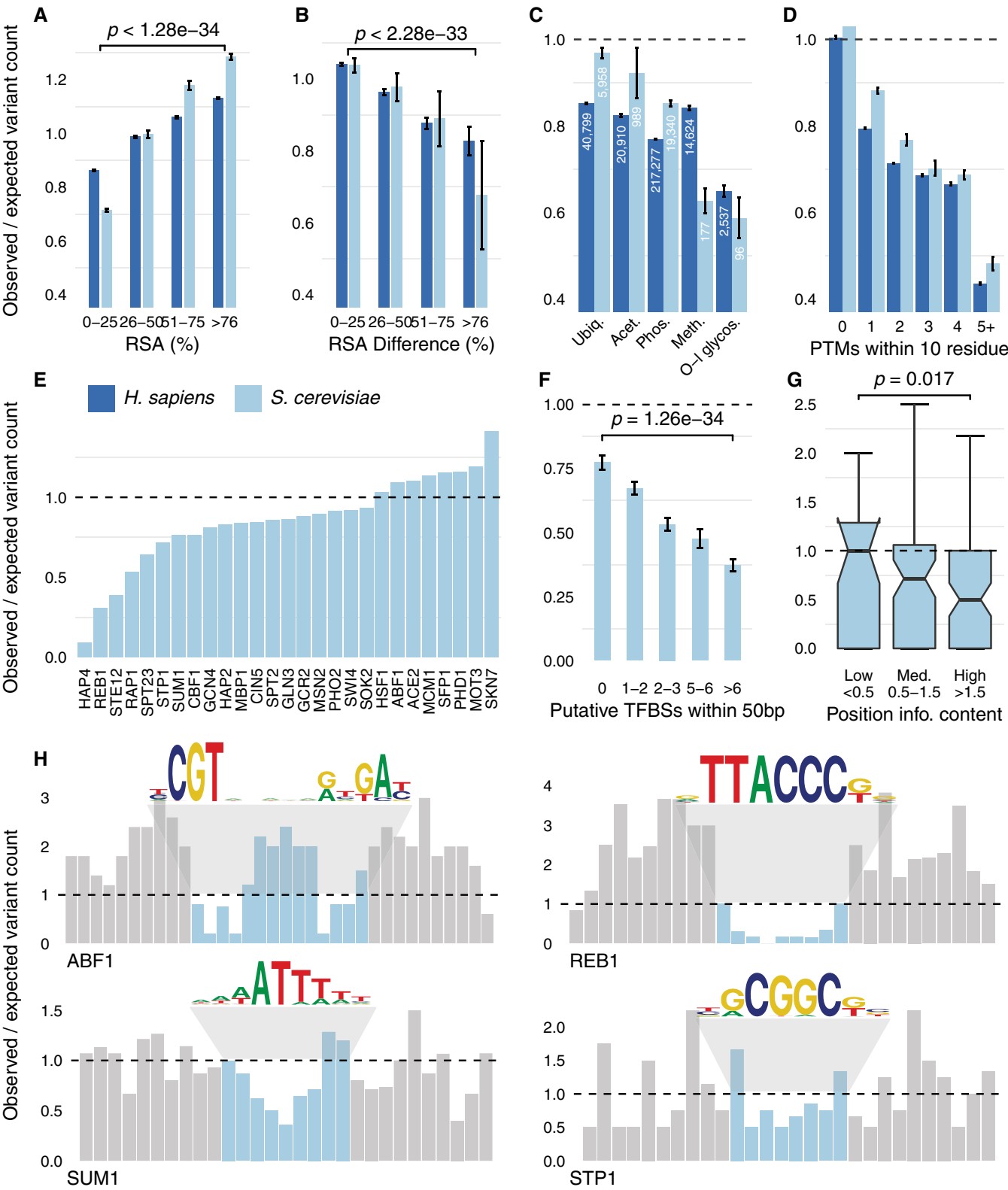

**Figure 1.**

(*n* = 212.2M *H. sapiens*, *n* = 53.4M yeast, *n* = 26.1M *E. coli*). To measure the impact on protein stability, the FoldX algorithm (Schymkowitz *et al*, 2005) was applied to 11,771 structures (including homology models) across the three organisms (Materials and

Methods and Fig EV1) and precomputed effects of 55.9M protein coding substitutions (*n* = 42.7M *H. sapiens*, *n* = 5.3M *S. cerevisiae*, *n* = 8.1M *E. coli*). We identified interface residues in 10,675 structures of binary PPIs from Interactome3D across the three organisms

**Figure 1. Population-level sequence constraint in genome functional elements.**

The level of sequence constraint was estimated using a ratio of the counts of genome variants across individuals of yeast and human compared with a random control region for different functional elements.

A  Regions buried within a protein structure with a low RSA typically exhibit higher evolutionary constraint.

B  Similarly, regions buried within interaction interfaces exhibit a high ΔRSA and demonstrate stronger sequence constraints.

C  Sequence constraint on PTMs, where numbers reflect the number of PTM sites for each modification.

D  PTMs with a higher number of neighbouring PTMs show stronger constraint.

E  Variability in constraint among bindings sites for TFs with at least 40 sites.

F  TFBSs that coexist with other binding sites are under stronger constraint.

G  Position-specific constraint shows that positions of higher relevance for binding in TFs with at least 20 sites are under stronger constraint. Notches represent the 95% CI in the median, box limits the IQR and upper whiskers the 75th percentile. The horizontal line represents the null expectation of no difference between observed and expected, same as in all other panels of this figure.

H  Four examples where the bar plots reflect the position-specific constraint in (blue) and around (grey) the binding site, along with sequence logos for the binding specificities.

Data information: (A, B, F) P-values represent a one-sided Wilcoxon test. (A, B, C, D, F) Error bars represent the standard deviation. One hundred random samples were used. (G) P-value shown is computed using a one-sided Kolmogorov–Smirnov test.

and similarly applied FoldX to compute the effects of 11.2M possible interface mutations on binding stability ($n$ = 7.2M $H.\ sapiens$, $n$ = 2.3M $S.\ cerevisiae$, $n$ = 1.6M $E.\ coli$). To identify variants that could impact kinase–substrate sites, we used MIMP (Wagih $et\ al$, 2015) to predict the impact of all possible 541,161 variants ($n$ = 485,736 $H.\ sapiens$, $n$ = 55,425 $S.\ cerevisiae$) falling within ±5 residues of a known kinase–substrate phosphorylation site (phosphosite) on a kinase's specificity. Specificities for 56 kinases in $H.\ sapiens$ and 46 kinases in $S.\ cerevisiae$ were considered. Kinase–phosphosite relationships for $E.\ coli$ are not well established and cannot be scored in the same way. For all other PTMs such as methylation, ubiquitination and acetylation for which we do not have explicit flanking sequence specificity models, a variant was considered damaging if it directly altered the modified site. This resulted in a total of 6.3M possible variants that could alter such PTM sites across the three organisms ($n$ = 5.8M $H.\ sapiens$, $n$ = 537,434 $S.\ cerevisiae$, $n$ = 9,177 $E.\ coli$). For linear motif information, not available for $E.\ coli$, we gathered 1,668 experimentally identified linear motifs ($n$ = 1,525 $H.\ sapiens$, $n$ = 143 $S.\ cerevisiae$), along with their derived regular expression pattern from the ELM database (Dinkel $et\ al$, 2012) and computed the impact of all possible 226,920 variants ($n$ = 205,120 $H.\ sapiens$, $n$ = 21,800 $S.\ cerevisiae$) on binding patterns. Finally, for TFBSs, for organisms without well-defined functional TFBSs ($H.\ sapiens$ and $S.\ cerevisiae$), we defined putative TF-gene regulatory network using TF-knockdown expression data and/or ChIP-seq/ChIP-chip (Materials and Methods). We then used PWMs to identify putative binding sites, and predict the impact (Materials and Methods) of all possible 3.6M variant substitutions ($n$ = 3.3M $H.\ sapiens$, $n$ = 236,382 yeast, $n$ = 46,768 $E.\ coli$) on specificities of 217 TFs ($n$ = 72 $H.\ sapiens$, $n$ = 104 $S.\ cerevisiae$, $n$ = 41 $E.\ coli$).

These precomputed variant effect predictions constitute a resource that can be used in diverse ways. In the next sections, we benchmark this resource and illustrate some of its possible applications.

**Functionally important positions are enriched in predicted deleterious variants**

In order to benchmark the variant effect predictions that underlie the mutfunc resource, we first asked whether essential genes would harbour fewer natural variants that are predicted to be deleterious. Essential genes in yeast (Giaever & Nislow, 2014) and human (Blomen $et\ al$, 2015) consistently demonstrated significantly lower

frequencies of variants predicted to affect conserved sites (SIFT score < 0.05, $P = 6.46 \times 10^{-17}$ human, $P = 3.64 \times 10^{-24}$ yeast, Fig 2B) and protein stability (ΔΔG pred > 2, $P = 1.58 \times 10^{-10}$ human, $P = 4.7 \times 10^{-4}$ yeast, Fig 2B). Variants of higher allele frequency in the population are expected to be less impactful, and in accordance with this, we observed an increase in deleterious scores, as predicted from SIFT and FoldX, for variants of lower allele frequencies (Fig 2C). In addition to allele frequencies, we analysed mutations that are known to be deleterious. For $H.\ sapiens$, we used 34,600 variants annotated to be pathogenic ($n$ = 17,167) or benign ($n$ = 17,433) from the ClinVar (Landrum $et\ al$, 2014). For $S.\ cerevisiae$, we used 8,083 variants consolidate by Jelier $et\ al$ (2011) as either tolerated ($n$ = 5,271) or affecting function ($n$ = 2,812; Materials and Methods). The different predictors consistently discriminated tolerated from pathogenic variants as measured by the area under the receiver operating characteristic curve (AUC). SIFT performed the best at discriminating pathogenic variants from benign (AUC $H.\ sapiens$ = 0.87, $S.\ cerevisiae$ = 0.92), followed by FoldX interfaces (AUC $H.\ sapiens$ = 0.64, $S.\ cerevisiae$ = 0.72) and FoldX stability (AUC $H.\ sapiens$ = 0.70, $S.\ cerevisiae$ = 0.62, Fig 2D).

For other heuristic-based predictors such as SLiMs, PTMs or stop gains/losses, we compared the proportion of pathogenic versus benign variants that disrupt or not the annotation. Despite the low number of pathogenic variants overlapping with these features, we observed an enrichment of pathogenic variants for mutations that disrupt such features (Fig 2E and F). The only exceptions were for PTM-disrupting variants in human and for linear motif-disrupting variants in yeast. In contrast, there were significant differences for the enrichment of pathogenic variants disrupting human linear motifs ($P = 5.23 \times 10^{-3}$) and yeast PTM sites ($P = 8.44 \times 10^{-7}$). For some of annotations, the lack of statistical significance may be due to the small number of testable variants.

The results here demonstrate that the predictors used in mutfunc are generally capable of enriching for variants of functional significance. The resource can be used to prioritize variants according to the degree of pathogenicity as well as provide molecular mechanisms affected.

**Predicting mechanistic impacts of variants of uncertain significance**

Variants that have been identified through disease-related genetic testing but are yet to be deemed benign or pathogenic are termed

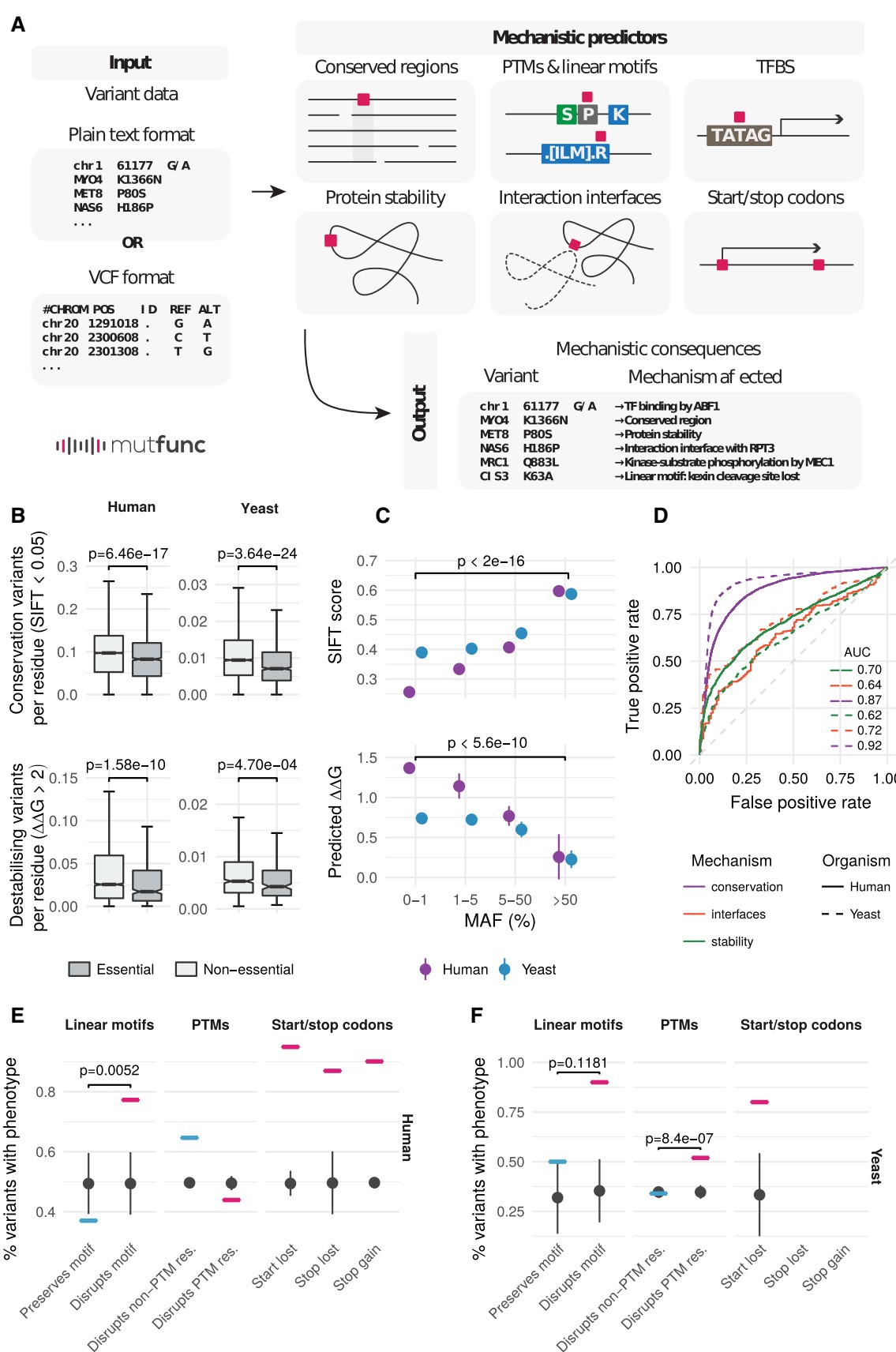

**Figure 2.**

**Figure 2.  The mutfunc resource and benchmarking of underlying variant effect predictors.**

A   The mutfunc interface provides an intuitive, user-friendly way by which users can query the resource using DNA or protein substitutions provided in plain text format or the variant call format (VCF). The impact of variants across different mechanisms is provided with information on impact strength in downloadable format and/or protein structural views.

B   The fraction of variants predicted to affect a conserved or structural important residues for essential and nonessential genes. For yeast SIFT, the number of essential/non-essential genes are 3,967 and 906, respectively. For yeast foldx the numbers are 925 and 281. For human sift the numbers are 15,542 and 1,575. For human foldx the numbers are 3,702 and 499.

C   Mean SIFT scores and predicted ΔΔG values for human and yeast variants within different MAF bins. Error bars represent the standard error, and *P*-values are calculated based on a one-sided Wilcoxon test.

D   Pathogenic and benign variants were obtained for human (from ClinVar) and yeast (curated) as described in the Materials and Methods section. These were used to benchmark the capacity of different predictors to discriminate between known pathogenic and benign variants.

E, F   The proportion of pathogenic versus benign variants that disrupt or not different functional annotations (SLiMs, PTMs or stop gains/losses) in human (E) and yeast (F). Number of replicates is 100 (i.e. random samples).

Data information: (B, E, F) *P*-values represent a one-sided Wilcoxon test. (E, F) Error bars for random samples represent the standard deviation.

variants of uncertain significance (VUS). The interpretation of such variants is a common challenge in genetics, one that is often aided by computational predictors. A total of 64,692 variants labelled with "uncertain significance" were collected from ClinVar (Landrum *et al*, 2014). VUSs were annotated using mutfunc and 21,584 variants were predicted impactful by at least one of the mechanistic predictors, not including SIFT ($n = 7,547$ stability, $n = 751$ interfaces, $n = 139$ linear motifs, 2,372 PTMs, 57 kinase binding). From these, we focused on variants predicted to impact the structural integrity of proteins (stability and interaction interfaces) since they hold the highest coverage.

Of the VUSs predicted to interfere with interface or protein stability, we retained those in which (i) the protein also harbours a known pathogenic variant with the same predicted structural impact, and (ii) both the pathogenic variant and VUSs are identified in patients with the same disease. This allows us to connect a variant of uncertain significance with a pathogenic variant by the fact that they occur in patients of the same disease and are predicted to have the same molecular consequence at the protein level. We demonstrate a few examples of VUSs that are predicted to alter binding (Fig 3A–C) or structural stability (Fig 3D and E). For instance, primary hyperoxaluria is a disease caused primarily by mutations in GRHPR, a glyoxylate and hydroxypyruvate reductase (Cramer *et al*, 1999; Cregeen *et al*, 2003), and its enzymatic activity requires homodimerization (Booth *et al*, 2006). For this enzyme, the variants R302H and E113K have been implicated in primary hyperoxaluria, are annotated to be pathogenic in ClinVar (Landrum *et al*, 2014) and are predicted here to impact on binding stability (Fig 3A, ΔΔG > 2.15). We can reason that other variants in patients of the same disease impacting on GRHPR homodimerization are therefore also likely to have the same phenotypic outcome. For example, the variant R171H is predicted to impact a conserved region as well as the homodimerization stability (Fig 3A, ΔΔG = 2.19, s < 0.018) and found in primary hyperoxaluria patients. Although R171H is of uncertain significance, our analysis strongly suggests that it is very likely to have the same phenotypic consequences and act via the same molecular mechanism as R302H and E113K. Similarly compelling examples are found for other proteins such as fumarate hydratase (Fig 3B) and lamin (Fig 3C).

Similar to interface variants, we analysed variants that destabilize the protein structure. We identified 1,182 VUSs predicted to alter stability in proteins containing pathogenic variants also predicted to be destabilizing. For instance, the ubiquitin ligase PARK2, implicated in Parkinson's disease, contains two variants (V56E and C232Y) annotated to be pathogenic in ClinVar (Landrum

*et al*, 2014) that we predict to impact on its stability. For this protein, two other variants of uncertain significance (R42H, V148E) were found in Parkinson's disease patients and also predicted to destabilize the protein (ΔΔG > 4.7, Fig 3D). Therefore, we would suggest that the R42H and V148E variants are likely to cause the same phenotypes as the V56E and C232Y variants. In the tumour suppressor serine/threonine-protein kinase STK11, pathogenic and VUS identified in Peutz–Jeghers syndrome patients can be similarly linked (Fig 3E).

The analysis here demonstrates how mutfunc could be applied to systematically prioritize pathogenic variants through altered mechanisms that may be the molecular cause of the phenotype and the combination of algorithms that cover different molecular mechanisms.

## *S. cerevisiae* strain genomic differences are a significant but weak predictor of phenotypic similarity

We sought to illustrate the use of mutfunc for genotype-to-phenotype association analysis. Using *S. cerevisiae* as a case study, we first phenotyped growth for a panel of 166 strains in 43 conditions (Materials and Methods). Colony sizes for strains were quantified, normalized and scored relative to all strains in a condition to produce a phenotypic measure defined as the S-score (Collins *et al*, 2006). Positive and negative values indicate higher or lower than expected growth for a given strain and a specific condition (Materials and Methods). S-scores for biological replicates demonstrated a high degree of concordance ($r = 0.91$, $P < 2.22 \times 10^{-16}$, Fig 4A) suggesting a high degree of confidence in phenotypic measurements. S-scores for each strain and growth condition are provided in Dataset EV1.

Hierarchical clustering of growth phenotypes revealed known clusters of related stressors (Fig 4B). Clusters of similar phenotypic profiles included, for example, UV light, cisplatin and MMS, which are all DNA-damaging agents (mean Pearson's $r = 0.51$); nystatin and caspofungin (Kathiravan *et al*, 2012), known to interfere with the cell wall ($r = 0.49$); and caffeine and rapamycin (Reinke *et al*, 2006), both known to inhibit TOR signalling ($r = 0.41$). Furthermore, strains belonging to the same population structure (Strope *et al*, 2015) or environmental origin often showed similar phenotypic profiles (Fig 4B). Genome sequences were available for 93 of the 166 profiled strains and used to calculate pairwise genomic similarity as the euclidean distance of the vector of SNPs. As expected, genetic similarity is significantly correlated with phenotypic similarity (Fig 4C, $r = 0.12$, $P < 0.0001$) but alone explains a small amount

**Figure 3.  Analysis of variants of uncertain clinical significance using mutfunc.**

A–C   Three examples of interaction interfaces containing variants predicted to impact binding stability. Subunits of the interaction complex are coloured in dark grey and white, and respective interface residues in dark green and green.

D, E   Two examples of variants predicted to impact protein stability. Pathogenic variants are labelled "P" in red, and VUSs "U" in blue.

of the phenotypic diversity. This is not unexpected since most genetic variation is neutral and distantly related strains accumulate variation that may not have an impact on the phenotypes tested. While strains having very similar genomes tend to have very similar phenotypes, strains with more divergent genotypes can show either a similar or very different phenotypic profiles (Fig 4C).

## Gene and complex disruption scores for genotype-to-phenotype associations

Given that most variants are expected to be neutral, we used the predictions collected in mutfunc to interpret the observed variants

in each strain at the gene level by computing a total gene burden or disruption score using the mechanistic predictions for conservation (SIFT), protein stability (FoldX) and protein truncating variants (PTVs, including start loss, nonstop and nonsense variants; Fig 5A). Scores produced by predictors are standardized to reflect the likelihood they are deleterious (Fig 5B, Materials and Methods). This allows for effects of rare variants to be combined across different protein positions and predictors into a single probability that the gene is affected ($P_{AF}$ score or burden score; Jelier *et al*, 2011; Galardini *et al*, 2017).

Using the gene-level disruption scores, we performed phenotype association analysis. Scores were binned based on high

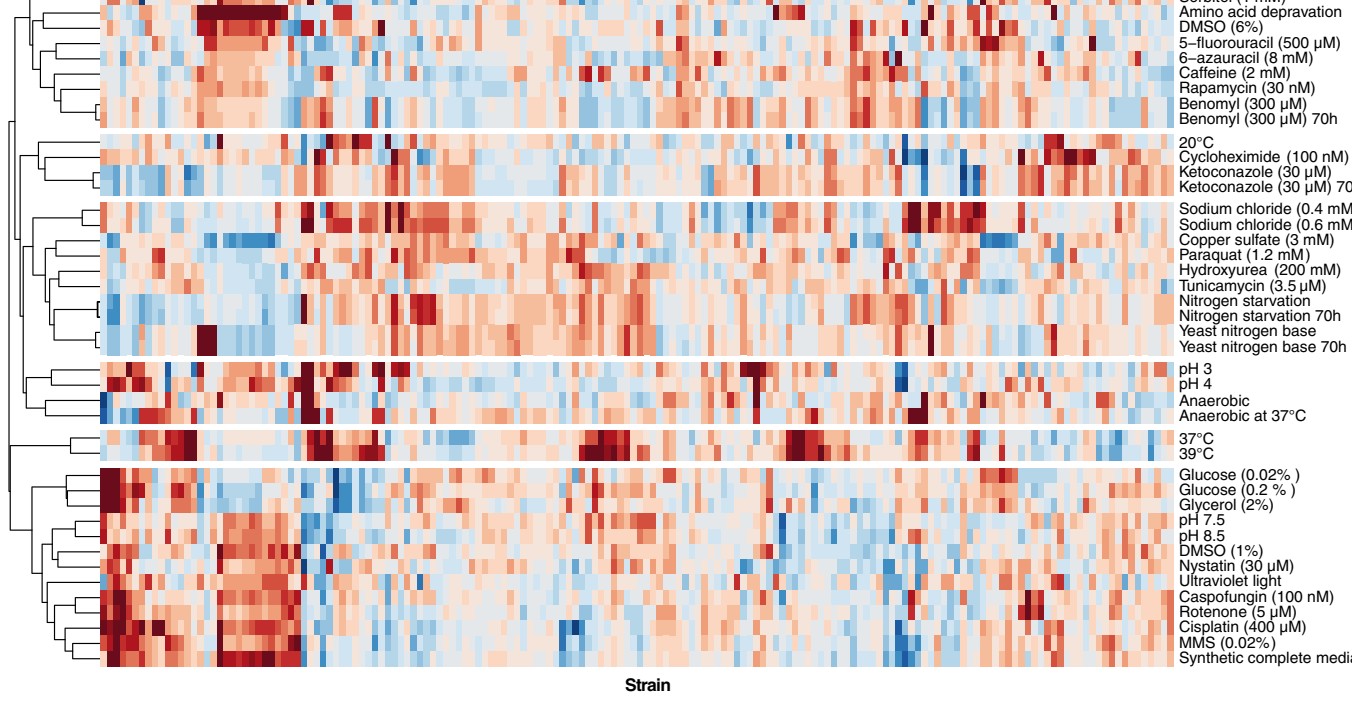

**Figure 4. Phenotypic screening of 166 yeast strains.**

A   Concordance between replicate s-score measurements.

B   Heatmap of s-scores showing hierarchical clustering of both strains and conditions reveals clusters of phenotypically similar strains and conditions.

C   Comparison of pairwise genotype and phenotype distances between 93 sequenced strains shows little observable correlation.

($P_{AF} > 0.90$) or low ($P_{AF} < 0.90$) burden (Fig 5C). Associations were carried out for 1,446 genes (with at least three strains containing a $P_{AF} > 0.90$) against growth phenotypes across 43 conditions (Materials and Methods). We identified 872 statistically significant gene–phenotype associations at $P < 1 \times 10^{-3}$ and FDR < 10%, with 82% (717/872) being negative. A negative association here indicates that the disruption of the gene is linked to decreased growth, while a positive association would suggest that the disruption is associated with a better than expected growth. Under the assumption that gene function is conserved across strains of *S. cerevisiae*, we expected these associations to be enriched in genes that cause a condition-specific phenotype when knocked out. Such association between gene KOs and condition-specific growth phenotypes exists for the laboratory strain as part

of extensive published chemical genetic studies. We found KO chemical genetic data for 35 of the 43 conditions tested. Of the significant negative associations, only 9% (65/717) are validated by the chemical genetic data. The validation rate increases for

higher effect sizes (Fig EV2A) to 15% (55/367) and 28% (20/71) for at Δ > 1 and at Δ > 1.8, respectively. However, based on permutation testing only the enrichment found at large effect sizes (Δ > 1.8) was significant (*P* = 0.04, Fig EV2A).

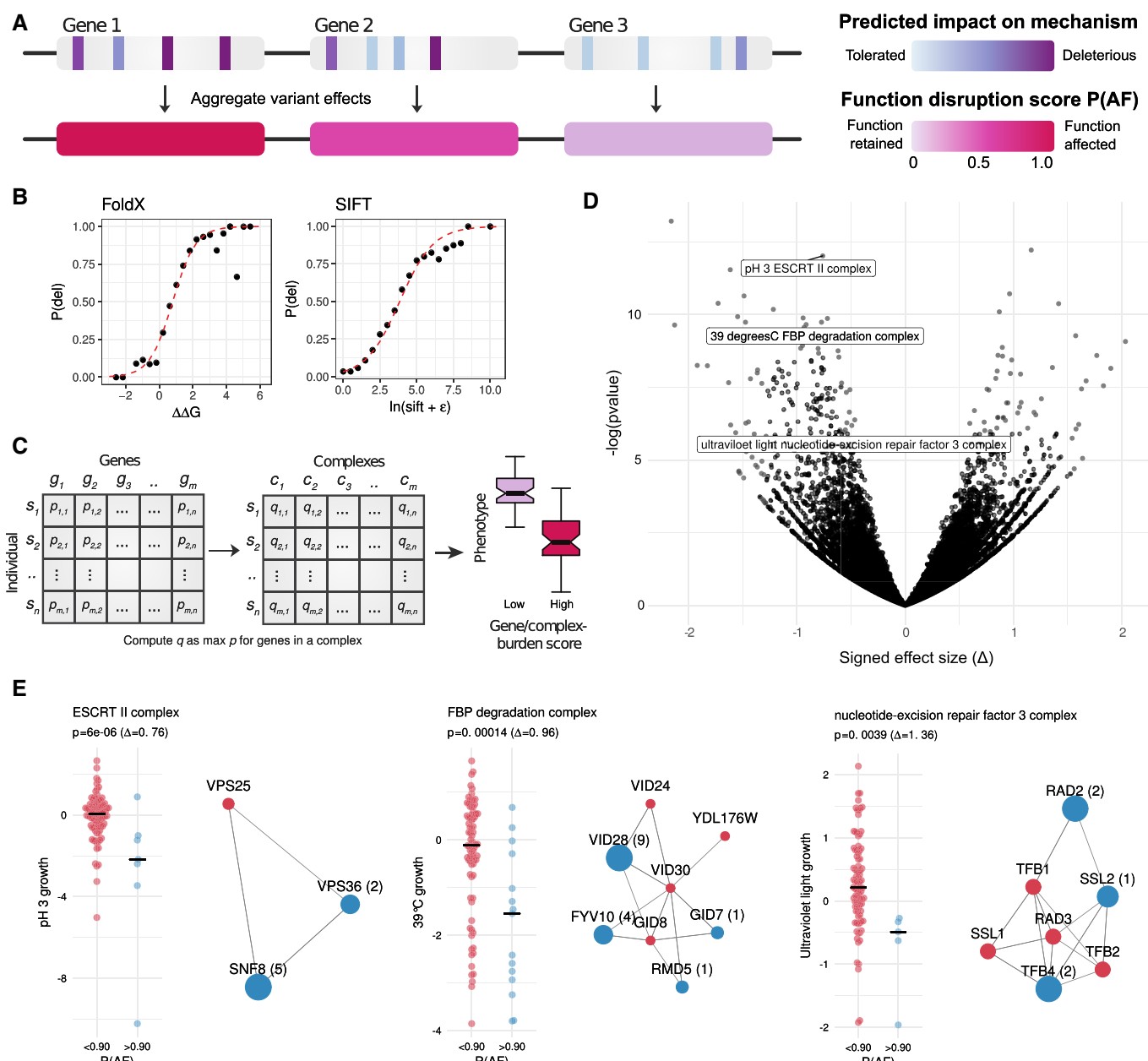

**Figure 5. Gene and protein complex-level aggregation of variant effects for phenotype association analysis.**

A  Diagram demonstrating the aggregation of variant impact. Each variant is first assigned a probability of deleteriousness, which are aggregated at the gene level using the maximum impact.

B  The probability of deleteriousness for FoldX and SIFT was computed by assessing the proportion of deleterious variants in gold-standard data for FoldX and SIFT. A logistic regression model (red line) is fit to compute subsequent probabilities. Protein complex-level burden scores were taken to be the maximal burden for any complex member.

C  Gene and complex burden scores for each strain, gene/complex-phenotype associations were carried out.

D  Volcano plot with gene–complex associations highlighting the effect size and *P*-value of selected examples.

E  S-score growth distributions for strains having a low ($P_{AF}$ < 90, red) or high ($P_{AF}$ > 90, blue) burden scores for three selected complexes. The protein subunits of each complex are shown with affected subunits in blue with the number of strains in which the subunit is predicted to be impaired in parenthesis. Subunits in red are not predicted to be impaired in any strain.

We next reasoned that protein complex members often act as coherent functional units and that the dysfunction of complex subunits often elicits similar phenotypic outcomes (Collins *et al*, 2007). Therefore, we aggregated the gene-level scores to identify complexes that were potentially defective in a given strain with the assumption that the complex was defective if at least one subunit was predicted to be impaired ($P_{AF} < 0.90$). We performed protein complex-level associations focusing on 263 complexes predicted to be defective in more than two strains (Fig 5D). A total of 106 significant complex–phenotype associations were identified ($P < 1 \times 10^{-3}$, FDR < 10%), 80 (75%) of which had a high effect size ($\Delta > 1$). The 80 associations involved 31 conditions and were preferentially negative associations (65 of 80, 81%). As we did for the gene-condition associations, we benchmarked the complex-condition associations and found a significant enrichment in KO chemical genetic data (Fig EV2B) that was significantly higher than observed based on random permutation testing (Fig EV2B, $P = 0.04$). This enrichment is observed only for stringent cut-offs for defining gene-deletion phenotypes from the KO chemical genetic studies (Hillenmeyer *et al*, 2008).

Some examples illustrate how the analysis at the protein complex level may increase power for the identification of associations (Fig 5E). For example, we found validated associations between the ESCRT II complex and growth in low pH (Xu *et al*, 2004). We have also found an association between high heat and the GID FBP degradation complex that contains several subunits (VID24, GID8 and VID28) that cause increased sensitivity to heat when deleted (as annotated in www.yeastgenome.org). In addition, we recover also the well-established requirement for the nucleotide excision repair complex for growth under UV light (Prakash *et al*, 1993). From these three validated examples, only one subunit of the ESCRT II complex (SNF8) shows a significant gene-level burden association with the respective condition (low pH). The other two complexes would not have been associated based on gene-level burden scores likely due to insufficient recurrency of mutation at the gene level.

This association analysis indicates that there is value in combining effects of rare variants at the protein and protein complex level to perform association studies. Although the current study is limited due to the relative small number of strains studied, it illustrates how mutfunc can be applied to the study of diverse set of problems.

## Discussion

The mutfunc resource makes use of established variant predictors to precompute millions of variant effects across the reference genomes of *H. sapiens*, *S. cerevisiae* and *E. coli*. This resource is not a new variant effect predictor nor an attempt to create an integrated score. The predictors used and their performance have been previously described, but the large computational effort and the accompanying web service (mutfunc.com) constitute a resource that facilitates their use. Within mutfunc, conservation effects hold the highest coverage, (*H. sapiens* 98.6%, *S. cerevisiae* 87.9% and 96.1% *E. coli*) followed by stability (*H. sapiens* 18.9%, *S. cerevisiae* 7.9% and 30.1% *E. coli*) and interfaces (*H. sapiens* 2.20%, *S. cerevisiae* 2.84% and 4.45% *E. coli*). Other mechanisms like PTMs and TFBSs are likely to have lower coverage, but it is unclear at the moment

what would constitute 100% coverage for these features. As additional data become available, mutfunc will be updated to improve coverage and future work could expand the set of mechanisms studied such as drug or small-molecule binding sites, RNA-binding interfaces, among others. The effects of variants on molecular and cellular phenotypes are increasingly being probed directly by large-scale mutagenesis experiments (Fowler & Fields, 2014; Weile *et al*, 2017), which will likely result in improved variant effect prediction algorithms (Gray *et al*, 2018). The curation of such experimentally determined effects and the improved algorithms can be integrated in future iterations of mutfunc.

A strength of mutfunc lies in its large set of precomputed SNV effects allowing for genome-wide variants to be rapidly queried. However, within such a framework, combinatorial and potential epistatic effects cannot be precomputed due to a large number of possible combinations. Similarly, many other types of genetic variation such as copy number variations and indels (Chuzhanova *et al*, 2003; Beroukhim *et al*, 2010) have not be considered in mutfunc due to their complex structure. Lastly, many organisms in which genetic variation is commonly studied are not included in mutfunc. These include *Mus musculus*, *Drosophila melanogaster* and *Arabidopsis thaliana*, which contain an abundance of data and could be added in the future.

Understanding how disrupted cellular mechanisms propagate to changes in phenotypes is critical for variant interpretation. We show here how different variants can be integrated using effect predictors and protein complex annotations to perform genotype-to-phenotype associations for full genome sequences. In addition, we and others have also shown how prior knowledge of gene function and variant effect predictions can be used to predict growth differences of different strains of *S. cerevisiae* (Jelier *et al*, 2011) and *E. coli* (Galardini *et al*, 2017). These analyses illustrate ways to calculate gene burden scores across different effect predictors. We found a significant but limited overlap between the gene-condition associations derived here with those found in gene KO studies in the reference laboratory strain. This small overlap could be due to a number of reasons including errors in variant effect predictions; limited sample size for the associations (i.e. 93 strains); and epistatic interactions of variants and different protocols for fitness measurements. The effects of a genetic variation *in vivo* can be complex and depend on both genetic and environmental factors (Burga *et al*, 2011; Wray *et al*, 2013; Perez *et al*, 2017). Several studies have shown that many variants annotated as disease-causing or predicted as deleterious have been identified in healthy humans (Xue *et al*, 2012). In addition to these potential causes of error, it is assumed here that the loss of function of a given gene will have the same phenotypic consequence across individuals of the same species. The extent by which this assumption is true remains to be tested.

Despite the limitations discussed, given the growing number of efforts to sequence exome and genomes for panels of individuals, the incorporation of variant prioritization by different approaches into association analyses will become more prevalent. The mutfunc resource can provide such variant effect predictions with mechanistic annotations for three species. We illustrate how this resource can be applied in different scenarios, and given the architecture used, these analyses can be easily incorporated into large-scale full genome or exome sequencing efforts.

## Materials and Methods

### Genetic variant data collection

A total of 896,772 genetic variants occurring in for 405 haploid and diploid *S. cerevisiae* strains were collected from four studies (Bergström *et al*, 2014; Strope *et al*, 2015; Gallone *et al*, 2016; Zhu *et al*, 2016). All but one study by Strope *et al* provided processed variant calls in VCF format. Variants were called for the Strope *et al* study using the following pipeline. Raw reads were obtained from the ENA resource (Leinonen *et al*, 2011). Adapter sequences were removed using cutadapt v1.8.1, and reads were mapped to the *S. cerevisiae* genome version 64 using BWA-MEM v0.7.8 (https://arxiv.org/abs/1303.3997). Duplicate reads were discarded using Picard v1.96 (https://github.com/broadinstitute/picard), and reads were realigned using the GATK IndelRealigner v3.3 (McKenna *et al*, 2010). Base alignment qualities were computed using SAMtools v1.2 (Li *et al*, 2009), and variants were called using FreeBayes v0.9.21-15-g8a06a0b and the following parameters –no-complex, –genotype-qualities, –ploidy 1 and –theta 0.006. The VCF was filtered for calls with QUAL > 30, GQ > 30 and DP > 4. VCF for individual *S. cerevisiae* strains was combined, and coding variants were called using the predictCoding function of the VariantAnnotation R package (Obenchain *et al*, 2014).

A total of 3,198,692 coding variants in *H. sapiens* for over 65,000 individuals were collected from the ExAC consortium along with corresponding adjusted allele frequencies. Ensembl transcript positions were mapped to UniProt by performing Needleman–Wunsch global alignment of translated Ensembl transcript sequences against the UniProt sequence using the pairwiseAlignment function in the Biostrings R package. The mapping between Ensembl transcript IDs (v81) and UniProt accessions was obtained from the biomaRt R package (Smedley *et al*, 2015). In the case that multiple alleles mapped to the sample single amino acid substitution, the one with the highest adjusted allele frequency was retained.

A total of 139,167 variants were obtained from ClinVar. Only variants that did not match one of the following clinical significance terms were removed: "Benign", "Benign/Likely benign", "Likely benign", "Likely pathogenic", "Pathogenic/Likely pathogenic" and "Pathogenic". Variants with a review status of "no assertion criteria provided" were also removed, as those reflect variants that have been assigned clinical significance without any particular criteria. The final filtered set contained 39,597 variants. Of these variants, 44% were classified as pathogenic or likely pathogenic. For *S. cerevisiae*, a total of 8,083 manually curated variants were obtained from Jelier *et al* (2011), 34.5% (2,812) of which were labelled as deleterious. Variants were collected from a combination of the UniProt database (Apweiler *et al*, 2004), Protein Mutant Database (Kawabata *et al*, 1999), *Saccharomyces* Genome Database (Cherry *et al*, 2012) and mutations that are identified in essential genes (Liti *et al*, 2009). These variants were used for the benchmarking of the variant effect predictors in Fig 2.

### Essential genes

A total of 2,501 essential genes identified using gene trapping technology in two haploid *H. sapiens* cell lines KBM7 and HAP1 were obtained from Blomen *et al* (2015). These were further filtered for genes that were essential in both cell lines, for a total of 1,734 genes. A total of 1,156 essential genes in *S. cerevisiae* were obtained from the *Saccharomyces* Genome Deletion Project (Giaever *et al*, 2002).

### Predicting impact on protein stability and protein interaction interfaces

Experimentally determined structures were obtained from the Protein Data Bank (PDB). Large structures that did not have a corresponding PDB file were downloaded in mmCIF format and converted to PDBs using the PyMOL Python library v1.2r3pre (pymol.org). Mapping of coordinates from PDB to UniProt residues was derived from the SIFTS database (Velankar *et al*, 2013). Structures with a resolution above three angstroms were discarded, and a single representative structure maximizing the coverage of the protein was retained. Homology modelling was carried out for proteins with no experimentally determined structures using ModPipe version 2.2.0 (Pieper *et al*, 2009) and the following parameters: –hits_mode 1110 and –score_by_tsvmod OFF. For each protein, we excluded models with a ModPipe Protein Quality Score lower than 1.1 and then kept the model with the highest normalized DOPE score. Finally, we excluded residues with a residue-level DOPE score (rDOPE) greater than 0 as stability predictions for such residues are error prone (Fig EV1). Experimental and homology modelled structures for protein interactions were obtained from the Interactome3D database (Mosca *et al*, 2012). Relative solvent accessibility (RSA) for all residue atoms was computed using NACCESS for proteins individually, and in the interaction complex. Interface residues were defined as those with any change in RSA. All other calculation of RSA was carried out using FreeSASA v1.1 (Mitternacht, 2016).

The impact of variant on stability was computed using FoldX v.4.0 (Schymkowitz *et al*, 2005). All structures were first split by chain into individual PDB files and repaired using the RepairPDB command, with default parameters. The Pssm command is then used to predict ΔG with numberOfRuns=5. This performs the mutation multiple times with variable rotamer configurations, to ensure the algorithm achieves convergence. The average ΔG of all runs is computed, and the ΔΔG is computed as the difference between the wildtype and mutant. The impact of variants on interaction interfaces is measured similarly, with the exception of structures being provided in binary interaction, rather than individual chains.

### Predicting the impact of variants on PTMs and linear motifs

For *S. cerevisiae*, a total of 20,056 phosphosites and 2,219 kinase–substrate associations were obtained from the PhosphoGRID database (Sadowski *et al*, 2013). A total of 1,070 of other PTM sites were obtained from the dbPTM database (Lee *et al*, 2006). For *H. sapiens*, all PTM data, including that of phosphorylation and kinase–substrate associations, were obtained from PhosphoSitePlus (Hornbeck *et al*, 2012), for a total of 296,147 sites. For *E. coli*, a total of 483 PTM sites were obtained from dbPTM (Lee *et al*, 2006). Linear motif data for *S. cerevisiae* and *H. sapiens*, including annotated linear motif binding sites and regular expression patterns, were obtained from the ELM database (Dinkel *et al*, 2016).

Impact of variants on phosphosites and flanking regions was measured using the MIMP algorithm (Wagih *et al*, 2015), with

default parameters. For other PTMs, a variant was predicted to be impactful if it resulted in the change of the modified residue. For linear motifs, a variant was predicted to be impactful if it causes a loss of match for associated regular expression pattern.

### Predicting the functional impact of variants using conservation

All protein alignments were built against UniRef50 (Suzek *et al*, 2015), using the seqs_chosen_via_median_info.csh script in SIFT 5.1.1 (Ng & Henikoff, 2003). The siftr R package (https://github.com/omarwagih/siftr), an implementation of the SIFT algorithm, was used to generate SIFT scores with parameters ic_thresh=3.25 and residue_thresh=2.

### Transcription factor binding sites

A total of 177 *S. cerevisiae* TFs binding models were collected in form of a position frequency matrices (PFMs) from JASPAR (Sandelin *et al*, 2004) and converted to position weight matrices (PWMs) using the TFBSTools R package (Tan & Lenhard, 2016). PWMs were trimmed to eliminate consecutive stretches of low information content (< 0.2) on either terminus. To identify genes likely regulated by a particular TF, a combination of TF knockout expression and ChIP-chip experiments was used, as similarly described in Gonçalves *et al* (2017). Genome-wide gene expression profiles for 837 gene-knockout strains were obtained from three studies (Chua *et al*, 2006; Hu *et al*, 2007; Kemmeren *et al*, 2014), 148 of which were a known TF with a defined PWM. Studies provided either a Z-score or *P*-value for each gene as a measure of over or under-expression, relative to the distribution of values for all genes. Two-tailed *P*-values were computed from Z-scores when a *P*-value was not provided. In cases where TF knockout was repeated between studies, the lowest *P*-value for each gene was used. ChIP-chip tracks for 355 TFs were collected from four studies (Harbison *et al*, 2004; Tachibana *et al*, 2005; Rhee & Pugh, 2011; Venters *et al*, 2011) via the *Saccharomyces* genome database. Of the 355 of the TFs, 144 (56%) had a defined PWM. Potential binding sites were then only searched for in TF-gene pairs with a *P*-value below 0.01 and the corresponding ChIP-chip region upstream of the regulated gene. A normalized log score of 0.80 was used as the cut-off for defining putative binding sites. Similarly, for *H. sapiens*, 454 TF PWMs were generated from JASPAR PFMs. ENCODE clustered ChIP-seq data were obtained for 161 TFs, of which 72 had a PWM. Only those regions were scored against the corresponding PWM. For *E. coli*, a total of 1,905 TF-matching sequences across 84 TFs were obtained from RegulonDB (Gama-Castro *et al*, 2016) and used to construct PWMs. A total of 2,416 experimentally identified TFBS were obtained for 79/84 TFs from RegulonDB. These sites were used as putative binding sites for downstream variant predictions.

Potential target sequences were scored against the PWM using the log-scoring scheme defined in Wasserman and Sandelin (2004) and normalized to the best and worst matching sequence to the PWM. The resulting score lies between 0 and 1, where 1 signifies strong predicted binding by the factor, whereas 0 signifies predicted lack of binding. Potential binding sites were scored in the presence ($S_{wt}$) and absence ($S_{mt}$) of a variant. Three separate metrics are used to quantify the change in binding between the reference and alternate allele. The first one is simply the difference in the normalized

log score, $S_{wt} - S_{mt}$, where a large positive value indicates loss of binding. The second is the difference in binding percentile. Here, random oligonucleotides are used to generate a negative distribution of log normalized scores for each TF. The percentile of each wild-type $p_{wt}$ and mutant scores $p_{mt}$ is computed from this distribution, and the difference, $p_{wt} - p_{mt}$, is used to quantify the magnitude of impact. The last is the difference in the relative information content. This can be thought of as the difference of letter height in a sequence logo. Given that the wildtype and mutant bases have relative frequencies of $f_{wt}$ and $f_{mt}$, respectively, and a position has an IC value of $\gamma$, then this is computed as $(f_{wt} \cdot \gamma) - (f_{mt} \cdot \gamma)$. This value ranges from 0 to 2, where 0 indicates little to no impact on a critical base, and 2 indicates a strong one.

### Implementation of mutfunc

Described predictors were used to precompute effects for all amino acid and nucleotide substitutions. The mutfunc web server at http://mutfunc.com uses the Java and Scala-based Play Framework v1.3.7 backend (http://www.playframework.org) along with a MySQL database. The front end utilizes a modified version of the Twitter Bootstrap UI library (http://getbootstrap.com/). Visualization tools used include a modified version of the neXtProt feature viewer v0.1.52 (https://github.com/calipho-sib/feature-viewer) for interactive visualization of protein sequence features, WebGL protein viewer v1.1 for interactive visualization of protein structures v1.8.1 (https://github.com/biasmv/pv) and a modified version of the JSAV v.1.10 library (https://github.com/AndrewCRMartin/JSAV) for visualization of multiple sequence alignments.

### Chemical genetic screening

The screening was carried out in 1,536 format on synthetic complete media with the addition of the appropriate chemical at a specific concentration. The Singer RoToR (Singer Instruments, UK) was used to replicate screening plates in 1,536 format. Agar plates were pinned onto the conditioned media and allowed to grow for 48 or 72 h at 30°C (unless specified otherwise). Each experiment was replicated once for quality control. After incubation, plates were imaged and colony sizes were extracted using IRIS version v0.9.7 (Kritikos *et al*, 2017) with the "Colony growth" profile, which extracts colony size, circularity and opacity from each colony in each plate. Individual strains were scored using the E-MAP software, which transforms colony sizes into s-scores (Collins *et al*, 2006). In brief, a surface correction algorithm is applied to each plate, the outer frame effect is corrected by bringing the two outer-most rows and columns to the plate middle median. All the plates are then normalized to the overall median, followed by a variance correction and finally the s-score calculation. The resulting s-scores are quantile normalized in each condition separately, and final s-scores from both replicates are averaged.

### Calculating gene and complex disruption scores

Scores produced by different predictors were standardized in order to reflect the likelihood of identifying a deleterious mutation ($P_{del}$). For SIFT, a curated gold-standard set of 8,083 variants in 1,346 yeast genes with known tolerated or deleterious effects were obtained

from Jelier *et al* (2011). The negative natural logarithm of the SIFT score was binned by 0.5, and for each bin, the proportion of deleterious variants was computed. A binomial logistic regression was fit to the proportion values and used to compute subsequent $P_{del}$ values for subsequent SIFT scores. For FoldX, 964 gold-standard mutations across 34 experimentally identified proteins structures with both experimentally quantified $\Delta\Delta G$ values and FoldX-predicted $\Delta\Delta G$ values were obtained from Guerois *et al* (2002). A variant was labelled destabilizing if $\Delta\Delta G$ was > 1. Mutations were binned by predicted $\Delta\Delta G$ at intervals of 0.4, and for each bin, the proportion of destabilizing variants was computed. A binomial logistic regression model was similarly fit to the data and used to compute subsequent $P_{del}$ for FoldX-predicted $\Delta\Delta G$ values. For variants disrupting start or stop codons, we assigned $P_{del}$ value of 1. Since nonsense variants occurring closer to the C-terminal of a protein are less likely to impact function, we only assign $P_{del}$ value of 1 for nonsense variants occurring in the first 50% of the protein; otherwise, a value of 0 was used. Gene burden scores are then computed as the variant with the maximum $P_{del}$ score and described the predicted likelihood that a protein has an affected function ($P_{AF}$). Similarly, for protein complexes the maximum $P_{del}$ score for any complex subunit was selected to reflect the protein complex $P_{AF}$ score. Variants with a MAF > 20% were considered unlikely to be deleterious given their high frequency in the population and were discarded prior to the burden score analysis.

**Genotype-to-phenotype association analysis**

The associations were carried out using the MatrixEQTL R package (Shabalin, 2012) with the modelLINEAR mode. The significance of the association was measured using a *t*-statistic. For the associations, genes and complex binarized $P_{AF}$ scores were used as genotypes where a $P_{AF}$ score above or below 0.9 is given a value 1 and 0, respectively, and growth phenotypes are used in lieu of gene expression. A *P*-value threshold of 0.001 was used for all associations, and multiple testing correction was carried out using the false discovery method. Effect size was computed using Glass's $\Delta$. For the case (*P*) and control (*n*) group, differences in the mean were computed relative to the standard deviation of one of the groups. Given the mean ($\mu_i$) and standard deviation ($\sigma_i$) for a given group *i*, this is computed as $\Delta_i = (\mu_P - \mu_n)/\sigma_i$. For robustness, this was computed in both direction and the final effect size, $\Delta$, is reported as the minimum absolute value of effect sizes in both directions.

# Data availability

The precomputed impact of single nucleotide variants in human, yeast and *E. coli* is available through the mutfunc web interface (www.mutfunc.com). Bug reports and feature requests can be submitted through the project's bug tracker at https://github.com/evocellnet/mutfunc-webserver.

**Expanded View** for this article is available online.

## Author contributions
OW designed, developed and implemented the mutfunc resource. OW, MG and DM contributed to data analysis. BB designed and performed the

*S. cerevisiae* strain growth experiments. AT and PB oversaw the project. OW, MG and PB wrote the manuscript, and all authors revised it.

## Conflict of interest
The authors declare that they have no conflict of interest.

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
