## [Review Process File · Molecular Systems Biology]

A resource of variant effect predictions of single nucleotide variants in model organisms

Omar Wagih, Marco Galardini, Bede Busby, Danish Memon, Athanasios Typas and Pedro Beltrao.

Review timeline:

Submission date:	2 nd May 2018
Editorial Decision:	13 th June 2018
Revision received:	29 th August 2018
Editorial Decision:	23 rd October 2018
Revision received:	19 th November 2018
Accepted:	21 st November 2018

Editor: Maria Polychronidou

Transaction Report:

1st Editorial Decision

13th June 2018

Thank you again for submitting your work to Molecular Systems Biology. We have now heard back from the three referees who agreed to evaluate your study. As you will see below, the reviewers think that the presented resource seems useful. They raise however a series of concerns, which we would ask you to address in a major revision.

During our referee cross-commenting process (in which the referees get a chance to anonymously comment on each other's reports) all reviewers agreed with the point raised by reviewer #2 that a comparison with existing tools will add value to the study. Regarding the comment of reviewer #3 on re-analyzing all yeast variant calls, reviewers #1 and #2 mentioned that in their opinion this would be a major undertaking, requiring a substantial amount of additional work. Reviewer #3 replied that s/he was concerned about the robustness of the paper in absence of such analyses, but suggested the following alternative solution, which we would ask you to consider in your revision: "The authors could make some comparison of the SNPs from the published studies to SNPs called simultaneously from all strains; they could either do this themselves without rerunning all their data, or they could compare to other studies that have done this (as in a recent paper by Sardi et al. PLoS Genetics, analyzing analyzed ~65 strains). They should also add a statement that the mapping results could be influenced by errors or differences in SNP calling that was done separately for different groups of strains."

All other issues raised need to be convincingly addressed. Please let me know in case you would like to discuss further any of the comments of the reviewers.

REFeree REPORTS.

Reviewer #1:

In this very interesting work, Wagih and colleagues describe a tool for predicting the effect of mutations from the perspective of protein structure. Although methods for predicting the severity of impact of variants (e.g. Ensembl VEP) are available, mutfunc offers fine-grained information about the effect of mutations in a structural context which was not previously easily available. Outside of the protein-coding regions, mutfunc offers additional, functionality of reporting whether transcription factor binding sites or start/stop codons are affected. The authors validate mutfunc by applying it to several datasets and demonstrate that it has predictive power. Overall, we believe this an excellent manuscript and will be a useful tool that will be an asset to the scientific community and we strongly support publication of this work in MSB.

Major comments:

A major point is the effect of homology modelling on the accuracy of mutfunc. Due to the limitations of homology modelling methods, prediction of the effect of mutations on the basis of modelled structures is expected to perform less well than in cases where a crystal structure is available. As there appears to be no model filtering based on quality scores, this effect will likely be particularly strong in the case of poor quality models. Some benchmarking of this will be valuable addition to the paper. This could be addressed in a variety of ways but perhaps the simplest would be to show predictive power for variants mapping to homology models separately from those mapping to crystal structures. It would also be useful to show the distribution of model quality (DOPE) scores. A related consideration is whether homology models outperform excluded protein structures with resolution $>3\text{\AA}$. Additionally, it would be helpful if information about homology model quality was displayed in the protein viewer so that a user is informed.

A second major point is that we were not able to obtain predictions for some regions of proteins that are covered by protein structures. For example, the highest resolution structure for yeast is 4UYR (Flo11). This structure covers e.g. residue G30 but we were not able to obtain predictions for this site. Similarly, we were unable to obtain predictions for e.g. human Smoothed homolog where structures with resolution $< 3\text{\AA}$ exist. I am wondering why these regions are missing. Correcting this issue or an explanation will be helpful. A more minor comment related to this is that the error message returned by mutfunc given is very generic ("Error: no results for variants entered, please try a different set of variants or select a different organism") and is the same regardless of the source of the problem. It would be much more helpful to know if the protein ID was not recognised, whether the amino acid given as reference doesn't match the database, if the queried position is not covered by any structures, etc.

Other minor comments:

More care should be given to the parsing of input. Currently, for start codons it is possible to enter the same amino-acid as both reference and alternative ("M1M") without any error message. Additionally, mutfunc occasionally appears to get stuck in a state where it will reject any given input as incorrect. After refreshing the website, the problem goes away.

It is unfortunate that HGNC gene symbols are not accepted as input. They are commonly used and very intuitive identifiers.

"Interactome3D database [23399932]." appears to be an error in referencing.

"Potential target sequences were scored against the PWM using the log-scoring scheme defined in (Wasserman & Sandelin, 2004)" has the wrong reference format.

"Within mutfunc, conservation effects hold the highest coverage, (H. sapiens 98.6%, S. cerevisiae 87.9%, and 96.1% E. coli) followed by stability (H. sapiens 18.9%, S. cerevisiae 16.9%, and 49.2% E. coli) and interfaces (H. sapiens 2.20%, S. cerevisiae 2.84%, and 4.45% E. coli). Other mechanisms like PTMs and TFBSs have lower coverage." In the case of TFBS and PTMs it's not clear what would constitute 100% coverage.

Reviewer #2:

Summary

In their present manuscript, the authors describe 'mutfunc', a tool for the prediction of the mechanistic effects of genetic variation that bundles a number of pre-existing tools specialising on different mechanistic aspects. They evaluate the component tools' abilities as to detect damaging mutations; describe the potential mechanisms for a selected set of Variants of Uncertain Significance (VUS) and apply their tool to predict associations between protein complexes and growth phenotypes in yeast using a complex-wide burden approach to boost sensitivity.

The presented work appears very promising as a resource for variant interpretation, especially the reporting of (predicted) affected mechanisms. However the work lacks a thorough comparison against other existing tools, which needs to be addressed. A similar problem exists for the burden test analysis: While predictions are made as to the associations, a follow-up validation is missing. It is difficult to assess the manuscript's significance without these.

Major points:

1. Many variant effect prediction tools have been published previously. A thorough comparison and evaluation against at least the most commonly used of these seems very necessary. How does mutfunc's precision/recall tradeoff compare against that of PolyPhen-2, CADD, PROVEAN. In particular, how does the method fare in comparison to other federated/ensemble predictors such as REVEL (Ioannidis et al., Am J Hum Genet, 2016)?
2. Follow-up validation is described for gene-level burden associations. A follow-up validation for the complex-burden association test is needed. For example, for the predicted associated complexes in yeast, mutagenesis/deletion of other complex members could be performed to examine if such mutations recapitulate the predicted phenotypes, or at least some post-hoc computational validation should be described.
3. In the VUS classification analysis, the authors apply additional filter criteria to the output of mutfunc. If these filtering steps are necessary to find meaningful results, does that not speak against the utility of the raw mutfunc output for variant classification?
4. While mutfunc uses predictors for many types of mechanisms via which mutations can affect fitness, not all of them are captured. For example, effects on splicing, codon usage, expression levels, localization, immunopresentation, small molecule binding can all affect the downstream organismal phenotype. The authors may need to discuss in how far this affects the tool's reliability.
5. Conversely, many mechanistic effects are already implicitly reflected in conservation. It would be interesting to evaluate in how far each of the other mechanisms improve the tool's classification performance compared to conservation alone?
6. On page 3, in the first paragraph of the results section (and figure 1d/f) the authors argue that sequence features that occur in clusters are more impactful. Could this not also be explained by the fact that with an increasing size of a given set of variants, the probability that at least one of them is impactful also increases? As a control, one might calculate the null distribution of random sets of variants of the same size as the given cluster.
7. On page 6: "For this protein two rare VUSs (R42H, V148E) identified in Parkinson's disease patients are similarly predicted to destabilise the protein ($\Delta\Delta G > 4.7$, Figure 3d) and are therefore likely to be pathogenic." "Likely pathogenic" is freighted with meaning for clinicians, and is the sort of determination that should be made with care or at least with all applicable caveats. What is the accuracy of $\Delta\Delta G$ predictions in general and of these predictions in particular? How well does predicted $\Delta\Delta G$ separate pathogenic from benign (or from randomly-chosen rare variants, given annotated-benign variants are highly biased towards common variants which may differ systematically from benign rare variants) .

8. On page 13: "Variants with a MAF > 20% were considered unlikely to be deleterious given their high frequency in the population and were discarded prior to the burden score analysis." This is good, but analysis of ExAc has suggested that the proper MAF threshold for this is much lower, maybe MAF > 0.5%.

Minor points:

1. On page 1: "However, GWASs are typically limited in their ability to explain the underlying mechanism that is influenced by the variant in question." We suggest: "However, GWASs are typically limited in their ability to identify the causal variant at the associated locus, and further limited by the ability to explain the underlying mechanism that may be influenced by candidate causal variants."
2. On page 3: "ubiquitin -> ubiquitylation"
3. On page 7: It is confusing to say that all p-values were corrected for FDR but then give a P-value and FDR that don't match. Would be clearer to indicate P-values that are nominal after that statement, or better yet say that the FDR value corresponding to each P-value was calculated.
4. On page 7: Positive and negative association should be defined in the context of burden tests. Positive means trait and burden are positively correlated?
5. There are a fair number of grammatical errors throughout the manuscript, which may need to be proof-read more carefully.
6. Figure 4b may benefit from application of branch-order optimization (e.g. dendsort).
7. Figure 5, panels (d) and (e) do not have a caption text.
8. Figure 5 (e) is missing a color legend.
9. Figure S1a is missing an x-axis label. Also the y-axis apparently shows the log(p-value) although the text mentions FDR-correction, so a q-value may be more appropriate.

Reviewer #3:

In this study, Wagih et al. compile a database of previously published human and yeast variants from natural populations along with predictions of functional impacts made by a variety of available algorithms and methods. I like the approach and the goal, although I'm not sure that Mol Sys Bio is the right home for publishing the database.

Nonetheless, I had a few concerns with the datasets that gave me pause. There were several results that were odd to me that seem in conflict with past yeast studies - it wasn't until I got to the Methods that I realized that the variants for different batches of strains (published by different labs) were not reanalyzed but taken from different studies that used different approaches. I suspect that this could significantly contribute to some of those discordant results outlined below; regardless, the variants need to be called by the same methods for this database to be useful. The human variants were called by the same consortium, so I presume they were called with a standardized pipeline and consistent parameters (but if this is not the case, it is a red flag). In my opinion, the variant calling needs to be redone, at least for the yeast strains, using a single pipeline. I realize that is a lot of work for the authors, but I think it's important for the database to be useful.

I will leave it to the editors to decide the fit, but below are specific comments that would improve the manuscript.

1. I found the references lacking in several key places.

- Second paragraph on page 3 describes that there are fewer substitutions within TF PWM positions of high information content as if this is a new result, but there have been seminal papers studying the relationship between substitution rate and information content. Those need to be cited here.

- Page 6, second paragraph, "For instance, the ubiquitin ligase PARK2, implicated in Parkinson's disease, contains pathogenic variants predicted to impact on its stability." What is the reference here? The authors argue that two other variants in this region are also predicted to be deleterious and thus due to proximity to the pathogenic variant they are also likely to disrupt stability, but the pathogenic site they're comparing to is only predicted to impact stability. Aren't these all just predictions?

- Paragraph spanning pages 6-7 cites several relationships between stressors but no references - are these known relationships or just logical to the authors?

- Second to last Results paragraph gives examples of different complexes linked to different stresses, but there is not a reference to be found in this paragraph - are these validated connections or just logical to the authors?

2. Perhaps I missed it in the Methods, but how was the ROC curve done in Fig 2? It would be useful to know what the TP and FP sets were. I could not understand Fig 2e from the legend given, more detail should be given (including n and not just %) so the reader can interpret the figure.

3. I was initially unclear at the top of Page 6, "Of the VUSs predicted .. we retained those in which 1) the protein also harbors known pathogenic variant ... and 2) both the pathogenic variant and VUSs are identified ..." I initially thought they were looking at alleles with multiple polymorphisms, but I think they are aggregating alleles for a single protein score for the mapping. A clearer presentation here would be useful.

4. Another section that was unclear, "We performed protein complex level associations focusing on 263 complexes with at least two high burden genes across strains." I presume they are not looking at complexes in which at least two members are predicted defective in the same strain, but rather that they are collapsing information across a group of genes and predicting that the complex is defective in one way or another in individual strains? This section should be clarified.

5. After compiling the database and doing some computational validations, the authors then measure fitness of 166 strains grown in 43 different conditions and then attempt to map the phenotypic variation to genetic variation. But several features of their data disagree with other published studies.

First, "As expected, genetic similarity alone is a poor predictor of phenotypic response similarity (Figure 4c)." But this is in contrast to population genetic models and observed data in yeast (e.g. Warringer et al. papers, Kvitek et al., and other yeast phenotyping studies). Strains that are genetically similar are generally more phenotypically similar, so this statement is incorrect. The plot in Fig 4c is also surprising - I could not find anywhere a description of what distance was used for the genetic and phenotypic distances, this needs to be clearly stated.

Second, I was surprised that the strains from the same populations did not cluster together in Fig 4 based on phenotypic similarity, since this is observed in other high-throughput yeast phenotyping studies. I could not find what the similarity metric was used in the clustering, but one of the tightest cluster is a group of yellow "unknown" strains which is odd.

It is possible that the discordant results compared to other studies emerge because variants were called differently by different studies - thus the genetic distances are not properly captured, because the variants have some association with the study that called them. Otherwise, I don't see why the authors are getting different trends than population genetics predicts and other data shows.

6. Fig 5d and 5e have no legend so I could not interpret. I felt that the text on this section was also not convincing, perhaps because I couldn't interpret the figure but also because of the lack of references for what the authors present as their ground truth.

7. Was the *E. coli* data used for anything in this manuscript? Apologies if I missed it, but I didn't see *E. coli* mentioned in any of the figures or the results. Perhaps the data are in the database but not analyzed here?

Point-by-point response to the editor and referee questions (in black) with our responses marked in blue

Editor

Thank you again for submitting your work to Molecular Systems Biology. We have now heard back from the three referees who agreed to evaluate your study. As you will see below, the reviewers think that the presented resource seems useful. They raise however a series of concerns, which we would ask you to address in a major revision.

During our referee cross-commenting process (in which the referees get a chance to anonymously comment on each other's reports) all reviewers agreed with the point raised by reviewer #2 that a comparison with existing tools will add value to the study. Regarding the comment of reviewer #3 on re-analyzing all yeast variant calls, reviewers #1 and #2 mentioned that in their opinion this would be a major undertaking, requiring a substantial amount of additional work. Reviewer #3 replied that s/he was concerned about the robustness of the paper in absence of such analyses, but suggested the following alternative solution, which we would ask you to consider in your revision: "The authors could make some comparison of the SNPs from the published studies to SNPs called simultaneously from all strains; they could either do this themselves without rerunning all their data, or they could compare to other studies that have done this (as in a recent paper by Sardi et al. PLoS Genetics, analyzing analyzed ~65 strains). They should also add a statement that the mapping results could be influenced by errors or differences in SNP calling that was done separately for different groups of strains."

All other issues raised need to be convincingly addressed. Please let me know in case you would like to discuss further any of the comments of the reviewers.

Thank you for the clarification on these points. We have addressed all points of the reviewers below point-by-point but we would like to discuss here the issue of comparing mutfunc with existing tools as this was a point that was agreed with all reviewers in cross-commenting but that we have no real way of performing.

As we describe in the manuscript, mutfunc is not a new variant effect predictor and it is also not attempting to integrate the scores from multiple variant effector predictors. In mutfunc we provide pre-computed variant effect predictions from several well established tools and methods. The value of mutfunc is the extensive set of pre-computations that were derived with best practice approaches across the genomes of 3 species. In particular the structure based analysis performed with FoldX is, to our knowledge, done here for the first time in such large scale. The objective here is to

provide easy access to such computations to labs that would not have the expertise or compute infrastructure to do this type of work.

Given that the methods used to build mutfunc (e.g. FoldX, SIFT, MIMP) are all well established and published approaches we don't see how we can compare mutfunc itself with other methods. There have been extensive tests comparing variant effect predictors (Thiltgen et al. PLOS One 2012, Martelotto et al. Genome Biology 2014, Thusberg et al. HGV 2011) and we will, in the future, revise and update the ones we select for inclusion within mutfunc as newer methods become available. Based on this and given that the methods we have used have been benchmarked and compared with other methods we don't know what the reviewers here mean when they asked us to compare mutfunc itself with other approaches.

Regardless, given that there is some amount of misunderstanding we have added text in the discussion section to further clarify that mutfunc is not a new variant effect predictor nor do we intend to provide an integrated score. Instead we give the user easy access to well established tools and methods that cover a wide array of different variant effect predictors that report on different mechanisms. The key being this diversity of potential mechanistic outcomes.

Reviewer #1:

In this very interesting work, Wagih and colleagues describe a tool for predicting the effect of mutations from the perspective of protein structure. Although methods for predicting the severity of impact of variants (e.g. Ensembl VEP) are available, mutfunc offers fine-grained information about the effect of mutations in a structural context which was not previously easily available. Outside of the protein-coding regions, mutfunc offers additional, functionality of reporting whether transcription factor binding sites or start/stop codons are affected. The authors validate mutfunc by applying it to several datasets and demonstrate that it has predictive power. Overall, we believe this an excellent manuscript and will be a useful tool that will be an asset to the scientific community and we strongly support publication of this work in MSB.

Major comments:

A major point is the effect of homology modelling on the accuracy of mutfunc. Due to the limitations of homology modelling methods, prediction of the effect of mutations on the basis of modelled structures is expected to perform less well than in cases where a crystal structure is available. As there appears to be no model filtering based on quality

scores, this effect will likely be particularly strong in the case of poor quality models. Some benchmarking of this will be a valuable addition to the paper. This could be addressed in a variety of ways but perhaps the simplest would be to show predictive power for variants mapping to homology models separately from those mapping to crystal structures. It would also be useful to show the distribution of model quality (DOPE) scores. A related consideration is whether homology models outperform excluded protein structures with resolution $>3\text{\AA}$. Additionally, it would be helpful if information about homology model quality was displayed in the protein viewer so that a user is informed.

We fully agree with the comment raised by the reviewer. To address this we have compared the performance of FoldX based predictions for homology based models and empirical structures. As the reviewer points out, for homology models there are quality criteria that can be used to define models and regions within the models that may be of lower or higher quality. We already had filtered out homology models with poor overall quality scores as defined by the ModPipe Protein Quality Score (MPQS). Models with an MPQS of >1.1 were considered reliable. This was not initially described in the Methods section but was now updated to reflect this. In addition, we further tested the possibility of filtering specific regions within the homology models based on a per residue DOPE score (rDOPE).

We selected from Protherm mutations that had $\Delta\Delta G > 2$ kcal/mol as true positive mutations likely to be destabilizing and all other mutations as not destabilizing. We then compared the capacity to discriminate these two classes of mutations using FoldX predictions and different structural models (see figure below). For empirical structures we compared NMR and xray structures of different resolutions. Overall xray structures with $<3\text{\AA}$ resolution resulted in higher prediction power than using NMR or xray structures with greater than 3\AA resolution. This is the basis of our decision to exclude NMR and poor resolution structures.

For homology based models we compared the performance when we used residues with a residue level rDOPE score <0 with those for $\text{rDOPE} > 0.05$. We observed that selecting residues with $\text{rDOPE} < 0$ results in FoldX based predictions that are of even higher performance than those obtained for xray structures. Presumably because high rDOPE scores regions are often loop segments or other flexible regions where destabilization will be harder to predict even with an xray structure.

Based on these results we decided to exclude from mutfunc FoldX residue scores based on homology models with residue level rDOPE scores greater 0. These residues had been already excluded for human so these changes only impacted *S. cerevisiae* and *E. coli*. This led to the exclusion of 36.9% of residues with structural based predictions that had at least 1 predicted deleterious variant. The result of this benchmark was added to the manuscript as Supplementary Figure 1.

After the exclusion of these residues we re-did all of the analysis previously performed and updated the figures in the manuscript. No finding was impacted by this and there were only minor changes to the results presented in Figure 2, Figure 5 and previously named Supplementary Figure 1 (currently Supplementary Figure 2). The text was updated accordingly.

A second major point is that we were not able to obtain predictions for some regions of proteins that are covered by protein structures. For example, the highest resolution structure for yeast is 4UYR (Flo11). This structure covers e.g. residue G30 but we were not able to obtain predictions for this site. Similarly, we were unable to obtain predictions for e.g. human Smoothened homolog where structures with resolution < 3Å exist. I am wondering why these regions are missing. Correcting this issue or an explanation will be helpful. A more minor comment related to this is that the error message returned by mutfunc given is very generic ("Error: no results for variants entered, please try a different set of variants or select a different organism") and is the same regardless of the source of the problem. It would be much more helpful to know if

the protein ID was not recognised, whether the amino acid given as reference doesn't match the database, if the queried position is not covered by any structures, etc.

We thank the reviewer for the feedback on the webserver. We have checked these specific examples mentioned and we do have the corresponding models analyzed but we only store in the mutfunc webserver the information regarding the variants that have a predicted deleterious consequence. This is due to performance issues where storing all variant effect predictions would make the on-the-fly retrieval of information impractical. This relates also with the error messaging where improvements need to be made in regards to the error let the user know if the protein ID exists and if the positions queried exist but has not reported consequences. Unfortunately, this will require a significant change of the underlying database structure and outputs that will require significant development time. We have changed the error message to explicitly say that no deleterious variants were found and in the Help section we try to make it clearer that we only store information on variants predicted to be deleterious.

Other minor comments:

More care should be given to the parsing of input. Currently, for start codons it is possible to enter the same amino-acid as both reference and alternative ("M1M") without any error message. Additionally, mutfunc occasionally appears to get stuck in a state where it will reject any given input as incorrect. After refreshing the website, the problem goes away.

We have addressed the first concern raised here and ignore any variant that does not cause a change in sequence. The second concern was not possible to address without a way to reproduce the issue. We have set up bug tracking in github (<https://github.com/evocellnet/mutfunc-webserver>) and added a link to the help section so that users can more easily report these issues in a way that we can reproduce and follow up on.

It is unfortunate that HGNC gene symbols are not accepted as input. They are commonly used and very intuitive identifiers.

Mutfunc does support HGNC gene symbols. It is possible that, as with the question above, the specific ID/variant pairs tested were not predicted to be deleterious.

"Interactome3D database [23399932]." appears to be an error in referencing.

"Potential target sequences were scored against the PWM using the log-scoring scheme defined in (Wasserman & Sandelin, 2004)" has the wrong reference format.

We have corrected these errors.

"Within mutfunc, conservation effects hold the highest coverage, (H. sapiens 98.6%, S. cerevisiae 87.9%, and 96.1% E. coli) followed by stability (H. sapiens 18.9%, S. cerevisiae 16.9%, and 49.2% E. coli) and interfaces (H. sapiens 2.20%, S. cerevisiae 2.84%, and 4.45% E. coli). Other mechanisms like PTMs and TFBSs have lower coverage." In the case of TFBS and PTMs it's not clear what would constitute 100% coverage.

We agree completely with the reviewer on this point and have added a similar sentence to the fact that we would not know be able to estimate coverage for these.

Reviewer #2:

Summary

=====

In their present manuscript, the authors describe 'mutfunc', a tool for the prediction of the mechanistic effects of genetic variation that bundles a number of pre-existing tools specialising on different mechanistic aspects. They evaluate the component tools' abilities as to detect damaging mutations; describe the potential mechanisms for a selected set of Variants of Uncertain Significance (VUS) and apply their tool to predict associations between protein complexes and growth phenotypes in yeast using a complex-wide burden approach to boost sensitivity.

The presented work appears very promising as a resource for variant interpretation, especially the reporting of (predicted) affected mechanisms. However the work lacks a thorough comparison against other existing tools, which needs to be addressed. A similar problem exists for the burden test analysis: While predictions are made as to the associations, a follow-up validation is missing. It is difficult to assess the manuscript's significance without these.

Major points:

1. Many variant effect prediction tools have been published previously. A thorough comparison and evaluation against at least the most commonly used of these seems very necessary. How does mutfunc's precision/recall tradeoff compare against that of PolyPhen-2, CADD, PROVEAN. In particular, how does the method fare in comparison to other federated/ensemble predictors such as REVEL (Ioannidis et al., Am J Hum Genet, 2016)?

We have addressed this point at the beginning of this rebuttal but we will briefly repeat here again that it is not clear to us what the reviewer's concern here is and how we may deal with it. As previously stated in the response to the editor, mutfunc is not a new variant effect predictor and it is also not an integrated score predictor as REVEL. Mutfunc itself does not have a precision/recall tradeoff that can be compared. The VEPs used in mutfunc have been thoroughly compared by many other papers for those, such as SIFT and FoldX that have competitor versions. The value of mutfunc lies in the genome-wide computational prediction of variant effects across a large number of **effect types**. The key point being that it provides mechanist information such as if the variant is predicted to impact on protein stability, interfaces, PTMs and TF binding sites. For example, the REVEL tool suggested by the reviewer is a predictor that integrates the results into a single score of functional relevance. With REVEL the user would not be able to know how the variant is impacting on function.

We think having a single place where users can obtain such predictions for well established tools and approaches is very useful and in this manuscript we show some example applications of mutfunc. To attest to this, although mutfunc has not yet been published we have had an average of 200 users per month in the past 3 months, a value that that has been growing.

As in the comment to the editor, we again reiterate that we can't see how we would perform the requested analysis. We could in principle compare SIFT and FoldX with many other similar VEPs but this has been done extensively by others. We acknowledge that there are many other VEPs that we could have included in mutfunc and we are committed to revising the set of VEPs we use in future updates of mutfunc as more powerful VEPs are created.

We have added some additional text to the discussion to emphasize that mutfunc is neither a new variant effect predictor nor an integrated score for different VEPs.

2. Follow-up validation is described for gene-level burden associations. A follow-up validation for the complex-burden association test is needed. For example, for the predicted associated complexes in yeast, mutagenesis/deletion of other complex members could be performed to examine if such mutations recapitulate the predicted phenotypes, or at least some post-hoc computational validation should be described. We already performed the post-hoc computational validation suggested by the reviewer in the original submission. Perhaps this was not sufficiently clear. We have tried to clarify in the complex level analyses that we attempted to validate the associations using gene KO information tested under the same conditions. In both cases we see a

small but significant enrichment in previously described gene-condition or complex-condition associations (see Supplementary Figure 2).

3. In the VUS classification analysis, the authors apply additional filter criteria to the output of mutfunc. If these filtering steps are necessary to find meaningful results, does that not speak against the utility of the raw mutfunc output for variant classification?

On the contrary, the VUS classification analysis provides an example of how mutfunc can be used to take advantage of the fact that it provides scores for VEPs covering different mechanisms. As an application of this level of information the users can then filter the resulting output exactly in the way that we did for the VUS example. This would not have been possible using a tool like SIFT or PolyPhen-2 or REVEL. We used this exactly as an example of the strength of mutfunc. Given genomes from different patients (or different strains of *S. cerevisiae* or *E. coli*) the user can rapidly upload a VCF file with variants and identify those that target the same protein-protein interface with calculated $\Delta\Delta G$ and at the same time a metric of sequence constraint based on SIFT. Finding out that strains/patients with the same phenotype tend to have a common mechanism of protein disruption is something that we think is extremely valuable.

As an additional example of how this aspect of mutfunc could be used we could point to the several manuscripts studying recurrence of mutations at protein interfaces in cancer (Espinosa et al. PLOS One 2014, Kamburov et al. PNAS 2015, Porta-Pardo et al. PLOS Comp Bio 2015). In each case, the authors have had to obtain a catalog of interfaces and map mutations to these. In these studies the predicted impact of the mutations in the interface stability was not considered although one would expect this to be important. Mutfunc allows for such analyses to be conducted easily, obviating almost all of the work done in these studies.

4. While mutfunc uses predictors for many types of mechanisms via which mutations can affect fitness, not all of them are captured. For example, effects on splicing, codon usage, expression levels, localization, immunopresentation, small molecule binding can all affect the downstream organismal phenotype. The authors may need to discuss in how far this affects the tool's reliability.

We agree with the reviewer that while we capture many different mechanisms there are many that we also don't cover. We are excited that we have managed to cover so many with the current version of the tool and we will work to add more in the future. We don't think this affects the reliability of the tool in the sense that we provide results for individual VEPs independently. The lack of coverage for potential other VEPs is already discussed for several mechanisms that may be added in the future.

5. Conversely, many mechanistic effects are already implicitly reflected in conservation. It would be interesting to evaluate in how far each of the other mechanisms improve the tool's classification performance compared to conservation alone?

As described above we are not attempting to provide an integrated score to describe the functional relevance of a given position or the impact of single variant. Based on the other comments of the reviewer this appears to be key point where the reviewer's expectations are not in line with our efforts. Each VEP gives results independently and they are not integrated in the mutfunc webserver. Even if all impactful variants found by other approaches were covered by a conservation based analysis such as SIFT, this would not inform the mechanism by which the variant affects the protein. As previously stated, the strength of mutfunc is to provide users with a variety of mechanistic predictors so that the user would have a prediction not just that the variant may have an impact but also what the impact is - protein stability, a given interface, a PTM, etc. We have added text in the discussion section to further clarify that mutfunc is not a new variant effect predictor nor do we intend to provide an integrated score.

6. On page 3, in the first paragraph of the results section (and figure 1d/f) the authors argue that sequence features that occur in clusters are more impactful. Could this not also be explained by the the fact that with an increasing size of a given set of variants, the probability that at least one of them is impactful also increases? As a control, one might calculate the null distribution of random sets of variants of the same size as the given cluster.

In Figure 1 of the manuscript we attempt to show how the diverse set of features we have compiled for these genomes (e.g. protein interfaces, PTM positions, TF binding sites) show signs of evolutionary constraint. We used the number of observed variants in the feature divided by the number of expected variants as the measure of constraint. At this point in the manuscript we do not yet use the estimated impact of the variants only the counts. In figure 1d/f we show the ratio of the number of variants over expected is lower for regions with increasing number of PTMs or TF binding sites. The expectation already takes into account the size of the region as suggested by the reviewer in this comment.

7. On page 6: "For this protein two rare VUSs (R42H, V148E) identified in Parkinson's disease patients are similarly predicted to destabilise the protein ($\Delta\Delta G > 4.7$, Figure 3d) and are therefore likely to be pathogenic." "Likely pathogenic" is freighted with meaning for clinicians, and is the sort of determination that should be made with care or at least with all applicable caveats. What is the accuracy of $\Delta\Delta G$ predictions in general and of these predictions in particular? How well does predicted $\Delta\Delta G$ separate pathogenic from benign (or from randomly-chosen rare variants, given annotated-benign variants are

highly biased towards common variants which may differ systematically from benign rare variants).

We agree with the reviewer that the term "likely pathogenic" may be inappropriate since we did not test the variant experimentally and it has not been associated with the disease itself. The suggested analysis was previously carried out in the original submission. The $\Delta\Delta G$ predictions provided by FoldX distinguishes human pathogenic from benign variants with an AUC of 0.70 (see Figure 2D of the manuscript). However, here the association is further made because the VUS has the same predicted molecular outcome and found in patients of the same disease as a known pathogenic variant. We have revised the manuscript to describe these variants as predicted to have a similar phenotypic outcome to a pathogenic variant avoiding the use of the term "Likely pathogenic".

8. On page 13: "Variants with a MAF > 20% were considered unlikely to be deleterious given their high frequency in the population and were discarded prior to the burden score analysis." This is good, but analysis of ExAc has suggested that the proper MAF threshold for this is much lower, maybe MAF > 0.5%.

Given that we have a much lower number of yeast strains analyzed here as there are human individuals included in ExAC we considered it best to keep the >20% MAF threshold.

Minor points:

1. On page 1: "However, GWASs are typically limited in their ability to explain the underlying mechanism that is influenced by the variant in question." We suggest: "However, GWASs are typically limited in their ability to identify the causal variant at the associated locus, and further limited by the ability to explain the underlying mechanism that may be influenced by candidate causal variants."

We have made the suggested change.

2. On page 3: "ubiquitin -> ubiquitylation"

This has been changed accordingly.

3. On page 7: It is confusing to say that all p-values were corrected for FDR but then give a P-value and FDR that don't match. Would be clearer to indicate P-values that are nominal after that statement, or better yet say that the FDR value corresponding to each P-value was calculated.

We removed this sentence to avoid confusion as the p-value and FDR cut-offs are indicated.

4. On page 7: Positive and negative association should be defined in the context of burden tests. Positive means trait and burden are positively correlated?

A positive correlation indicates that the burden is associated with worse than expected growth whereas negative correlation indicates that high burden correlates with a better than expected growth phenotype. We have revised the text to make this point clearer.

5. There are a fair number of grammatical errors throughout the manuscript, which may need to be proof-read more carefully.

We have revised the text in attempt to remove any such errors.

6. Figure 4b may benefit from application of branch-order optimization (e.g. dendsort).

The hierarchical clustering output is meant only to provide a qualitative view of groups of strains and conditions. Quantitative claims on the expected similarity of related conditions and the relationship between genome similarity and phenotype similarity are made in the text.

7. Figure 5, panels (d) and (e) do not have a caption text.

8. Figure 5 (e) is missing a color legend.

9. Figure S1a is missing an x-axis label. Also the y-axis apparently shows the log(p-value) although the text mentions FDR-correction, so a q-value may be more appropriate.

We thank the reviewer for pointing out these errors. We have made the suggested corrections. The volcano plot in the previous figure S1a was removed given that we felt it was not required for the description of the results.

Reviewer #3:

In this study, Wagih et al. compile a database of previously published human and yeast variants from natural populations along with predictions of functional impacts made by a variety of available algorithms and methods. I like the approach and the goal, although I'm not sure that Mol Sys Bio is the right home for publishing the database.

We thank the reviewer for finding the approach and goal to be of interest. We emphasize that the pre-calculated variant effect predictions stored in mutfunc are not just for the natural variants that we studied, we pre-calculated all possible variant effect predictions across the genomes of the three species. Any researcher interested in

predictions for their own studies can easily provide a VCF file from a sequencing effort and obtain the same types of results as illustrated in our examples.

Nonetheless, I had a few concerns with the datasets that gave me pause. There were several results that were odd to me that seem in conflict with past yeast studies - it wasn't until I got to the Methods that I realized that the variants for different batches of strains (published by different labs) were not reanalyzed but taken from different studies that used different approaches. I suspect that this could significantly contribute to some of those discordant results outlined below; regardless, the variants need to be called by the same methods for this database to be useful. The human variants were called by the same consortium, so I presume they were called with a standardized pipeline and consistent parameters (but if this is not the case, it is a red flag). In my opinion, the variant calling needs to be redone, at least for the yeast strains, using a single pipeline. I realize that is a lot of work for the authors, but I think it's important for the database to be useful.

I will leave it to the editors to decide the fit, but below are specific comments that would improve the manuscript.

1. I found the references lacking in several key places.

- Second paragraph on page 3 describes that there are fewer substitutions within TF PWM positions of high information content as if this is a new result, but there have been seminal papers studying the relationship between substitution rate and information content. Those need to be cited here.

We agree with the reviewer and we had already mentioned the work of Spivakov et al. where this was discovered for human and drosophila TF binding sites. We had not intended to present this as a new finding in itself, but rather as a confirmation that our predicted TF binding sites for *S. cerevisiae* are biologically relevant as they show expected signs of constraint. We have revised the text to make it clearer that this has been previously shown for human/fly TF binding sites.

- Page 6, second paragraph, "For instance, the ubiquitin ligase PARK2, implicated in Parkinson's disease, contains pathogenic variants predicted to impact on its stability." What is the reference here? The authors argue that two other variants in this region are also predicted to be deleterious and thus due to proximity to the pathogenic variant they are also likely to disrupt stability, but the pathogenic site they're comparing to is only predicted to impact stability. Aren't these all just predictions?

The pathogenic variants are defined through ClinVar, which is based on literature curation and submissions from genetic testing centers. The fact that the VUS were

found in Parkinson's patients is also not a prediction. What is predicted is that the variants may share a similar molecular consequence that is to destabilize PARK2. We are suggesting that the prediction of a common molecular consequence for mutations is a useful feature in predicting variants with similar phenotypic consequences. It is not simply that the variants are predicted to have some impact but that they are predicted to have the same molecular outcome (i.e. disrupting an interface, stability, etc). We have revised this section of the text to better clarify what is known versus what is predicted.

- Paragraph spanning pages 6-7 cites several relationships between stressors but no references - are these known relationships or just logical to the authors?

The mentioned examples are very well established relationships. We added citations for the relationships that are more specific to yeast.

- Second to last Results paragraph gives examples of different complexes linked to different stresses, but there is not a reference to be found in this paragraph - are these validated connections or just logical to the authors?

We have added citations for the validations of the selected complex-condition associations.

2. Perhaps I missed it in the Methods, but how was the ROC curve done in Fig 2? It would be useful to know what the TP and FP sets were. I could not understand Fig 2e from the legend given, more detail should be given (including n and not just %) so the reader can interpret the figure.

We agree with the reviewer that this information was not clear in the manuscript. For human we used 39,597 ClinVar variants annotated to be benign or pathogenic as the false positive and true positive sets. For *S. cerevisiae* we use a set of 8,083 mutations curated to have deleterious or neutral consequences. We have improved the description of the source of the variants in the figure legend and added a note in the corresponding methods section.

3. I was initially unclear at the top of Page 6, "Of the VUSs predicted .. we retained those in which 1) the protein also harbors known pathogenic variant ... and 2) both the pathogenic variant and VUSs are identified ..." I initially thought they were looking at alleles with multiple polymorphisms, but I think they are aggregating alleles for a single protein score for the mapping. A clearer presentation here would be useful.

For the analysis of variants of uncertain significance (VUS) we are not aggregating variants for a single protein score. We are identifying variants from different patients of the same disease that are predicted to have the same molecular consequence (e.g. impacting on the same protein interface). If a variant of uncertain significance is found in patients of the same disease as those caused by a known pathogenic variant and both

of these are predicted to have the same predicted molecular consequence then this variant of uncertain significance is more likely to also be pathogenic. We have revised this section to make it clearer.

4. Another section that was unclear, "We performed protein complex level associations focusing on 263 complexes with at least two high burden genes across strains." I presume they are not looking at complexes in which at least two members are predicted defective in the same strain, but rather that they are collapsing information across a group of genes and predicting that the complex is defective in one way or another in individual strains? This section should be clarified.

We agree with the reviewer that this description needs improving. Each complex in each strain is considered to be disrupted if at least one of the members of the complex is predicted to be disrupted. We disregarded any complex that was not predicted to be disrupted in more than 2 strains. We have revised the text accordingly.

5. After compiling the database and doing some computational validations, the authors then measure fitness of 166 strains grown in 43 different conditions and then attempt to map the phenotypic variation to genetic variation. But several features of their data disagree with other published studies.

First, "As expected, genetic similarity alone is a poor predictor of phenotypic response similarity (Figure 4c)." But this is in contrast to population genetic models and observed data in yeast (e.g. Warringer et al. papers, Kvitek et al., and other yeast phenotyping studies). Strains that are genetically similar are generally more phenotypically similar, so this statement is incorrect. The plot in Fig 4c is also surprising - I could not find anywhere a description of what distance was used for the genetic and phenotypic distances, this needs to be clearly stated.

We don't think that the observations here are in opposition with previous findings. We had not intended to claim that there is no significant relationship between genotype and phenotype. In fact, there is a significant correlation between these two distances (Fig 4C, $r = 0.12$, $p < 0.0001$). What we meant was that the genetic distance alone explains a small amount of the variation in phenotypes. We believe the observed results are quite intuitive - If two strains are genetically similar then they tend to have similar phenotypes but as the genetic distance increases also does the variation of the similarity of the phenotypes. Large amounts of variants across two strains do not necessarily predict that they will have strong differences in phenotypic responses because a large number of the variants are neutral. We have revised the text to make this point clearer.

The genetic distance metric for each pair of strains is the euclidean distance of the SNP vectors. We have added this information to the results.

Second, I was surprised that the strains from the same populations did not cluster together in Fig 4 based on phenotypic similarity, since this is observed in other high-throughput yeast phenotyping studies. I could not find what the similarity metric was used in the clustering, but one of the tightest cluster is a group of yellow "unknown" strains which is odd.

It is possible that the discordant results compared to other studies emerge because variants were called differently by different studies - thus the genetic distances are not properly captured, because the variants have some association with the study that called them. Otherwise, I don't see why the authors are getting different trends than population genetics predicts and other data shows.

As described above, we do not believe that our findings are in any way contradicting previous observations. We have revised the statements to make it clear that there is a significant correlation between genome distance and phenotypic distance but that the genome similarity explains very little of the phenotypic variation. While this manuscript was under review, a paper was published describing the genotypes and some phenotypes for a panel of over 1000 yeast strains (Peter *et al.* Nature 2018). To demonstrate that our observations are not due to differences in variant calling for different strains we used the genotype and phenotype data published in that study and calculated genotype similarity as the euclidean distances between SNP vectors. We found for this study a very similar correlation between genome distance and phenotype similarity for pairs of strains (see Figure below) as observed in ours. We see a highly significant correlation between the two distances but with a poor correlation coefficient.

Similarly, for our previous study of *E. coli* strains we again found a similar relationship between these genomic and phenotypic distances for pairs of *E. coli* strains (Galardini *et al.* eLife 2017).

In summary, we don't believe that our findings are influenced by differences in variant calling pipelines for the strains and they are not contradicting previous studies. We thank the reviewer for bringing this point up and we have changed the text to say that these two distances are significant but that genotype differences alone explain a small proportion of the genotypic differences.

6. Fig 5d and 5e have no legend so I could not interpret. I felt that the text on this section was also not convincing, perhaps because I couldn't interpret the figure but also because of the lack of references for what the authors present as their ground truth. We have corrected the figure legends and have now also provided with additional references to provide with additional context for the analysis of these results.

7. Was the *E. coli* data used for anything in this manuscript? Apologies if I missed it, but I didn't see *E. coli* mentioned in any of the figures or the results. Perhaps the data are in the database but not analyzed here?

The *E. coli* data was not used in this current manuscript but we have used it a parallel study (Galardini *et al.* eLife 2017). We think this *E. coli* study again further illustrates the usefulness of the mutfunc database.

Thank you for sending us your revised manuscript. We have now heard back from the two reviewers who were asked to evaluate your manuscript. As you will see below, reviewer #1 is satisfied with the performed revisions and is supportive of publication. However, reviewer #2 thinks that mutfunc needs to be directly compared with variant effect predictor tools, since many of its applications are related to examining variant effects. During our pre-decision cross-commenting process (in which the reviewers are given the chance to make additional comments, including on each other's reports), reviewer #1 came back to us with the following comments: "I am on the fence with what referee #2 is requesting. I think the referee has a point that explicit comparison will be useful (or required as this referee feels). The authors' response that this is not a new variant effect predictor is also understandable. The potential problem is that readers, depending on their background or awareness, might think that mutfunc is a variant effect predictor and might use it in that capacity incorrectly rather than use it for interpreting mechanistic effects. One possibility would be to make explicit comparisons as originally mentioned by the referees but there are too many methods to compare against and it may not be fair to the authors. The alternative is to explicitly mention that this is not a variant effect predictor but a tool for interpreting mechanistic effects and make this point very clear at multiple places in the manuscript and on the accompanying websites."

We would not be opposed to the inclusion of direct comparisons with variant effect predictor tools, in case you feel inclined to perform them. However, we think that such analyses are not mandatory for the acceptance of the study for publication. We would only ask you to perform text changes along the lines of the suggestions by reviewer #1, in order to make it clear to the reader that mutfunc is not a new variant effect predictor. We would also recommend slightly editing the title of the study to reflect the resource value of the study.

REFEREE REPORTS.

Reviewer #1:

We are very pleased to see that the authors implemented or addressed the changes were suggested. The difference in performance between high and low quality homology models is even more striking than we would have predicted. Because of this, it is clearly necessary to exclude low quality homology models and we believe the rDOPE score threshold chosen by the authors for excluding them is correct and improves the usefulness of mutfunc. As regards changes to input handling and error messages, we are satisfied with the authors' response and the improvements they have made.

In short, this is an important contribution and we would be happy for the manuscript to be published in present form.

Reviewer #2:

The authors address most of the points made in the initial review. However, a major point agreed upon by all three reviewers was that the authors should compare mutfunc's performance to other existing tools. The authors responded to this with the argument that mutfunc should not be compared with predictors like Polyphen-2, PROVEAN or REVEL because it "is not a new variant effect predictor" but rather a collection of pre-calculated mechanistic effects. Yet all of the applications for mutfunc that the authors showcase in the manuscript (VUS reclassification, and burden testing at gene and complex level) fall in the domain of variant effect prediction and could be performed with other tools. Thus we would argue that it would be of importance to any reader whether using mutfunc for the applications demonstrated in the manuscript is indeed their best possible option, or if mutfunc would be better used as a secondary tool to explore possible mechanistic explanations for variant effects that are predicted by other tools with potentially greater sensitivity or precision.

For example, in their application for VUS re-classification prioritization, the authors use a qualitative filtering approach to identify VUS which likely disrupt a mechanism that has been shown

to be disease causative for other variants in the same gene. This prioritization approach is indeed likely to enrich for variants that are also disease-causative and does indeed produce tempting mechanistic anecdotes, but the question remains if this enrichment is superior to simply prioritizing VUS with high Polyphen-2 or PROVEAN scores. To answer this question, a cross-validation using known variant classifications from Clinvar could be performed both for the authors' filtering approach and for Polyphen-2 cutoffs.

Similarly, the burden tests with respect to yeast growth phenotypes can and should be compared to using existing tools. The authors distill a single metric, the P_AF score, from the mutfunc predictions and use that metric to inform the burden on a gene or complex. If a burden were instead determined using e.g. Polyphen-2 scores, would this lead to a worse agreement with the knock-out phenotypes?

2nd Revision - authors' response

19th November 2018

Please find enclosed the revised manuscript with the new title "A resource of variant effect predictions of single nucleotide variants in model organisms". We made some changes to the text of the manuscript and website to emphasize the resource aspect of the work and have also attempted to address the editorial requests. Please find below a point-by-point response with our response is dark blue.

Thank you for sending us your revised manuscript. We have now heard back from the two reviewers who were asked to evaluate your manuscript. As you will see below, reviewer #1 is satisfied with the performed revisions and is supportive of publication. However, reviewer #2 thinks that mutfunc needs to be directly compared with variant effect predictor tools, since many of its applications are related to examining variant effects. During our pre-decision cross-commenting process (in which the reviewers are given the chance to make additional comments, including on each other's reports), reviewer #1 came back to us with the following comments: "I am on the fence with what referee #2 is requesting. I think the referee has a point that explicit comparison will be useful (or required as this referee feels). The authors' response that this is not a new variant effect predictor is also understandable. The potential problem is that readers, depending on their background or awareness, might think that mutfunc is a variant effect predictor and might use it in that capacity incorrectly rather than use it for interpreting mechanistic effects. One possibility would be to make explicit comparisons as originally mentioned by the referees but there are too many methods to compare against and it may not be fair to the authors. The alternative is to explicitly mention that this is not a variant effect predictor but a tool for interpreting mechanistic effects and make this point very clear at multiple places in the manuscript and on the accompanying websites."

We would not be opposed to the inclusion of direct comparisons with variant effect predictor tools, in case you feel inclined to perform them. However, we think that such analyses are not mandatory for the acceptance of the study for publication. We would only ask you to perform text changes along the lines of the suggestions by reviewer #1, in order to make it clear to the reader that mutfunc is not a new variant effect predictor. We would also recommend slightly editing the title of the study to reflect the resource value of the study.

We thank both reviewers for their constructive criticism. We have amended the manuscript and the mutfunc website to further emphasize the fact that mutfunc is not a predictor per se. As an example, the landing page of mutfunc is now reading "Precomputed mechanistic consequences of mutations", and we have added to the initial description the following sentence: "We have precomputed data for all possible mutations, using existing algorithms to allow a quick and efficient lookup". Further changes have also been made in the "help" page, adding the following text in the introduction: "The annotations/predictions are based on the computation on the impact of all possible variants using existing algorithms that cover different mechanisms listed below". As requested we have also changed the title to emphasize the resource nature of the work. Overall, we hope that these changes sufficiently stress the resource nature of mutfunc.

Pedro Beltrao
MSB-18-8430R**